# Phenotypic diversity and temporal variability in a bacterial signaling network revealed by single-cell FRET

**Johannes M Keegstra[1], Keita Kamino[1], François Anquez[1†], Milena D Lazova[1], Thierry Emonet[2,3], Thomas S Shimizu[1]***

[1]AMOLF Institute, Amsterdam, The Netherlands; [2]Department of Molecular, Cellular and Developmental Biology, Yale University, New Haven, United States; [3]Department of Physics, Yale University, New Haven, United States

**Abstract** We present *in vivo* single-cell FRET measurements in the *Escherichia coli* chemotaxis system that reveal pervasive signaling variability, both across cells in isogenic populations and within individual cells over time. We quantify cell-to-cell variability of adaptation, ligand response, as well as steady-state output level, and analyze the role of network design in shaping this diversity from gene expression noise. In the absence of changes in gene expression, we find that single cells demonstrate strong temporal fluctuations. We provide evidence that such signaling noise can arise from at least two sources: (i) stochastic activities of adaptation enzymes, and (ii) receptor-kinase dynamics in the absence of adaptation. We demonstrate that under certain conditions, (ii) can generate giant fluctuations that drive signaling activity of the entire cell into a stochastic two-state switching regime. Our findings underscore the importance of molecular noise, arising not only in gene expression but also in protein networks.
DOI: https://doi.org/10.7554/eLife.27455.001

**\*For correspondence:**
shimizu@amolf.nl

**Present address:** [†]Laboratoire de Physique des Lasers, Atomes et Molécules, UMR CNRS 8523, Universite Lille 1, France

**Competing interests:** The authors declare that no competing interests exist.

## Introduction

Cellular physiology is deeply shaped by molecular fluctuations, resulting in phenotypic diversity and temporal variability that can be both detrimental and beneficial (*Rao et al., 2002*; *Kussell and Leibler, 2005*; *Lestas et al., 2010*; *Hilfinger et al., 2016*). One of the most important and well-studied sources of intracellular fluctuations is stochastic gene expression (*Elowitz et al., 2002*; *Eldar and Elowitz, 2010*; *Raj and van Oudenaarden, 2008*), which can generate substantial cell-to-cell variability in protein levels within isogenic populations under invariant environmental conditions. Such heterogeneity in protein counts are readily measurable by fluorescent-protein reporters (*Elowitz et al., 2002*; *Ozbudak et al., 2002*) , but mechanistically tracing the consequences of such molecular noise to the level of complex cellular phenotypes such as signaling and motility remains a significant challenge, in part due to the multitude of interactions between gene products, but also because each of those interactions can, in principle, become an additional source of noise. In this paper, we study how multiple sources of molecular noise, arising in both gene expression and protein-protein interactions, affect performance of the *E. coli* chemotaxis network, a canonical signaling pathway.

In bacteria, gene-expression noise tends to manifest itself as stable cell-to-cell differences in phenotypes that persist over the cell's generation time, because typical protein lifetimes are longer than the cell cycle (*Li et al., 2014*). The architecture of signaling networks can have a profound influence on their sensitivity to such noise-induced differences in protein levels, and it has been shown that the design of the *E. coli* chemotaxis network confers robustness of a number of signaling parameters, such as precision of adaptation, against variability in gene expression (*Barkai and Leibler,*

**eLife digest** Many sophisticated computer programs use random number generators to help solve challenging problems. These problems range from achieving secure communication across the Internet to deciding how best to invest in the stock market. Much research in recent years has found that randomness is also widespread in living cells, where it is often called "noise". For example, the activity of some genes is so unpredictable to the extent that it appears random. Yet, relatively little is known about how such gene-expression noise propagates up to change how the cell behaves. Many open questions also remain about how cells might exploit these or other fluctuations to achieve complex tasks, like people use random number generators.

Bacteria perform a number of complex tasks. Some bacteria will swim toward chemicals that suggest a potential reward, such as food. Yet they swim away from chemicals that could lead them to harm. This ability is called chemotaxis and it relies on a network of interacting enzymes and other proteins that coordinates a bacterium's movements with the input from its senses.

Keegstra *et al.* set out to find sources of noise that might act as random number generators and help the bacterium *E. coli* to best perform chemotaxis. An improved version of a technique called *in vivo* Förster resonance energy transfer (or *in vivo* FRET for short) was used to give a detectable signal when two proteins involved in the chemotaxis network interacted inside a single bacterium. The experiments showed that this protein network amplifies gene-expression noise for some genes while lessening it for others. In addition, the interactions between proteins encoded by genes acted as an extra source of noise, even when gene-expression noise was eliminated.

Keegstra *et al.* found that the amount of signaling within the chemotaxis network, as measured by *in vivo* FRET, varied wildly over time. This revealed two sources of noise at the level of protein signaling. One was due to randomness in the activity of the enzymes involved in tuning the cell's sensitivity to changes in its environment. The other was due to protein interactions within a large complex that acts as the cell's sensor. Unexpectedly, this second source of noise under some conditions could be so strong that it flipped the output of the cell's signaling network back and forth between just two states: "on" and "off".

Together these findings uncover how signaling networks can not only amplify or lessen gene-expression noise, but can themselves become a source of random events. The new knowledge of how such random events interact with a complex trait in a living cell – namely chemotaxis – could aid future antimicrobial strategies, because many bacteria use chemotaxis to help them establish infections. More generally, the new insights about noise in protein networks could help engineers seeking to build synthetic biochemical networks or produce useful compounds in living cells.
DOI: https://doi.org/10.7554/eLife.27455.002

1997; *Kollmann et al., 2005*). On the other hand, cell-to-cell differences in behavior can also be advantageous for isogenic populations under uncertain and/or time-varying environments, and it has been argued that the manner in which the chemotaxis network filters gene expression noise to shape phenotype distributions could be under selective pressure (*Frankel et al., 2014*; *Waite et al., 2016*).

In principle, molecular noise arising in processes other than gene expression, such as protein-protein interactions within signaling pathways, can also contribute to cellular variability. However, such noise sources tend to be harder to study experimentally because, in contrast to gene-expression noise, which can be characterized by measuring fluorescent reporter levels (*Elowitz et al., 2002*; *Raser et al., 2004*), requirements for *in vivo* measurements of protein-protein interactions tend to be more demanding and no generically applicable strategies exist. The *E. coli* chemotaxis system provides a compelling experimental paradigm for addressing protein-signaling noise, because a powerful technique for *in vivo* measurements of protein signaling, based on Förster resonance energy transfer (FRET), has been successfully developed (*Sourjik and Berg, 2002a*; *Sourjik et al., 2007*).

The chemotaxis network controls the motile behavior of *E. coli*, a run-and-tumble random walk that is biased by the signaling network to achieve net migrations toward favorable directions. The molecular mechanisms underlying this pathway have been studied extensively (for recent reviews,

see refs. (*Wadhams and Armitage, 2004*; *Tu, 2013*; *Parkinson et al., 2015*)). In brief, transmembrane chemoreceptors bind to ligand molecules, inhibiting the autophosphorylation of a central kinase, CheA. When active, CheA transfers its phosphate to CheY to form CheY-P. Meanwhile, the phosphatase CheZ dephosphorylates CheY-P to limit the signal lifetime. CheY-P binds to a flagellar motor, which in turn increases the chance of the motor to turn clockwise, leading to a tumble. An adaptation module consisting of the enzymes CheR and CheB implements negative integral feedback by tuning the sensitivity of the chemoreceptors via reversible covalent modifications that restore the kinase activity (and CheY-P level).

Despite its relative simplicity, this pathway exhibits many interesting network-level functionalities, such as cooperative signal amplification (*Segall et al., 1986*; *Sourjik and Berg, 2002a*; *Bray et al., 1998*), sensory adaptation (*Barkai and Leibler, 1997*; *Alon et al., 1999*), and Weber's law and fold-change detection (*Mesibov et al., 1973*; *Lazova et al., 2011*; *Clausznitzer et al., 2014*), and FRET microscopy has proven extremely powerful in characterizing such signal processing of the chemotaxis pathway, especially in *E. coli* (*Sourjik and Berg, 2002a*; *Sourjik and Berg, 2004*; *Shimizu et al., 2010*; *Oleksiuk et al., 2011*), but also in *Salmonella* (*Lazova et al., 2012*; *Rosier and Lazova, 2016*) and *B. subtilis* (*Yang et al., 2015*). It has been implemented in various ways (*Sourjik and Berg, 2002a*; *Sourjik and Berg, 2002b*; *Shimizu et al., 2006*; *Kentner and Sourjik, 2009*; *Neumann et al., 2012*), but most commonly by using CFP and YFP as the FRET donor-acceptor pair, fused to CheY and CheZ, respectively. To date, however, nearly all applications of FRET in the bacterial chemotaxis system have been population-level measurements in which signals from hundreds to thousands of cells are integrated to achieve a high signal-to-noise ratio. A pioneering study applied FRET at the single-cell level to study spatial heterogeneities in CheY-CheZ interactions (*Vaknin and Berg, 2004*), but those measurements were limited to relatively short times due to phototoxicity and bleaching.

By exploring a range of fluorescent proteins as FRET pairs, and improving measurement protocols, we have developed a robust method for single-cell FRET measurements of chemotactic signaling dynamics in single bacteria over extended times. The data reveal extensive cell-to-cell variability, as well as temporal fluctuations that are masked in population-level FRET measurements. In contrast to previous single-cell experiments that relied on measurements of motor output or swimming behavior (*Berg and Brown, 1972*; *Spudich and Koshland, 1976*; *Segall et al., 1986*; *Korobkova et al., 2004*; *Park et al., 2010*; *Masson et al., 2012*), FRET alleviates the need to make indirect inferences about intracellular molecular interactions through the highly noisy 2-state switching of the flagellar motor, whose response function can vary over time due to adaptive remodeling (*Yuan et al., 2012*). In a typical experiment, we are able to obtain dozens of (up to ~100) single-cell FRET time series simultaneously, to efficiently collect statistics of phenotypic diversity and temporal variability.

## Results

### Single-cell FRET reveals pervasive phenotypic diversity in intracellular signaling

To measure variability in intracellular signaling, we adapted a FRET assay for chemotaxis widely used for population-level measurements with fluorescent fusions to CheY and its phosphatase CheZ (*Sourjik and Berg, 2002a*). On timescales longer than the relaxation of CheY's phosphorylation/dephosphorylation cycle, the FRET level reflects the phosphorylation rate of CheY by the CheA kinase, thus providing an efficient *in vivo* measurement of the network activity (*Figure 1—figure supplement 1*). Instead of the conventional CFP/YFP FRET pair we used the fluorophores YFP and mRFP1 to avoid excitation with blue light, which induces considerably stronger phototoxicity and also perturbs the chemotaxis system as a repellent stimulus (*Taylor and Koshland, 1975*; *Taylor et al., 1979*; *Wright et al., 2006*). Fusions of these fluorophores to CheZ and CheY still yield a fully functional phenotype (*Wolfe and Berg, 1989*), when observing chemotaxis on soft agar (see *Figure 1—figure supplement 1d*).

A field of *E. coli* cells expressing this FRET pair were immobilized on a glass surface imaged in two fluorescence channels, and segmented offline to obtain fluorescence intensities of donor and acceptor. From the fluorescence ratio, FRET time series for each cell in the field of view (see

Materials and methods) can be computed, after dividing out the decay (**Figure 1—figure supplement 1**) in each channel due to bleaching. Ratiometric FRET provides an anti-parallel response signature and confers robustness to parallel fluctuations that affect both fluorescent channels, such as differences in absolute fluorescence intensity due to inhomogeneous illumination and differences in cell size.

For wildtype cells (**Figure 1a**) we found that the ensemble mean of single-cell FRET responses, $\langle \mathrm{FRET} \rangle (t)$, agrees well with previous population-level measurements (**Sourjik and Berg, 2002a**). Upon prolonged stimulation with a saturating dose of attractant $\alpha$-methylaspartate (MeAsp), $\langle \mathrm{FRET} \rangle (t)$ rapidly fell to zero before gradually returning to the pre-stimulus level due to adaptation. Upon removal of attractant, $\langle \mathrm{FRET} \rangle (t)$ rapidly increased to a maximum before returning to the pre-stimulus baseline. Single-cell FRET time series, $\mathrm{FRET}_i(t)$, had qualitatively similar profiles, but the kinetics of adaptation and response amplitudes demonstrate differences from cell to cell. For each cell, $\mathrm{FRET}_i(t)$ is limited by the autophosphorylation rate of CheA and hence is proportional to $a_i[\mathrm{CheA}]_{\mathrm{T},i}$ (provided [CheY] and [CheZ] are sufficiently high, see Materials and methods), in which $a_i$ is the activity per kinase ($0 \leq a_i \leq 1$) and $[\mathrm{CheA}]_{\mathrm{T},i}$ the total concentration of receptor-kinase complex of the $i$-th cell. The FRET level of each cell is thus bounded at a value $\mathrm{FRET}_{i,\mathrm{max}}$ which occurs when its kinases are fully active ($a_i$=1), and can be measured by the removal of a sufficiently large stimulus after adaptation (as in the experiment of **Figure 1**). Hence from $\mathrm{FRET}_i(t)$ the activity per kinase $a_i(t)$ can be readily determined by normalizing each FRET time series by its maximum response $a_i(t) = \mathrm{FRET}_i(t)/\mathrm{FRET}_{i,\mathrm{max}}$ (**Figure 1b**). The steady-state activity $a_{0,i}$, defined as the time-average of $a_i(t)$ before the addition of attractant, was found to vary from cell-to-cell with a coefficient of variation $CV(a_0) = 0.23$ (**Figure 1c**). The network activity controls the flagellar motor rotation, and hence this is consistent with the observation that cells in an isogenic population exhibit a broad range of steady-state tumble frequencies (**Spudich and Koshland, 1976**; **Bai et al., 2013**; **Dufour et al., 2016**).

The adaptation precision is defined as its post-adaptational activity level divided by the pre-stimulus level ($\Pi = a_{\mathrm{adapted},i}/a_{0,i}$), hence a precision of 1 refers to perfect adaptation. The adaptation kinetics are quantified by the recovery time $\tau_{\mathrm{recovery}}$, the time required for each cell to recover to 50% of its post-adaptation activity level ($a_{\mathrm{adapted},i}$). When observing the distributions of these parameters we noted that the cell-to-cell variability is high in the precision $\Pi$ (**Figure 1d**, $CV$=0.40) but the average precision (0.79) agrees well with population measurements (**Neumann et al., 2014**). The variation is also substantial in $\tau_{\mathrm{recovery}}$ (**Figure 1e**, $CV$=0.20) considering that the underlying kinetics of receptor methylation (catalyzed by CheR) involve thousands of events per cell, but falls within the range of ~20-50% from previous reports in which single-cell recovery times were estimated from motor-rotation or swimming-behavior measurements (**Berg and Tedesco, 1975**; **Spudich and Koshland, 1976**; **Min et al., 2012**). The time required to recover from a saturating amount of attractant is determined not only by the stimulus size, but also the methylation rate of receptor modification sites catalyzed by CheR and the number of such sites that need to be methylated. Variability in the recovery time is thus likely to reflect cell-to-cell variability in the ratio between the expression level of CheR and that of the chemoreceptor species responding to ligand (Tar for the experiment in **Figure 1**).

The diversity we observed here in adaptation precision, recovery time and steady-state activity was not explained by variation in salient experimental parameters (**Figure 1—figure supplement 2a–f**), are reproducible across experimental days (**Figure 1—figure supplement 2g**), and, on average, agree well with previous population-level FRET experiments and single-cell flagellar-based experiments. We thus conclude that single-cell FRET allows efficient measurement of signaling dynamics within individual bacteria to reveal variability in a wide variety of signaling parameters.

## Diversity in the ligand response is modulated during population growth

The chemoreceptor clusters in *E. coli* are the central processing units and are responsible for signal integration and amplification. The sensory output of the cluster, the activity of the kinase CheA, is activated by a mixture of chemoreceptors. Cooperative interactions within the receptor-kinase complex leads to amplifications of small input stimuli and weighting different input signals. It has been shown that the composition of the receptor-kinase complexes can affect both the amplification as well as the weighting of different input signals (**Ames et al., 2002**; **Sourjik and Berg, 2004**;

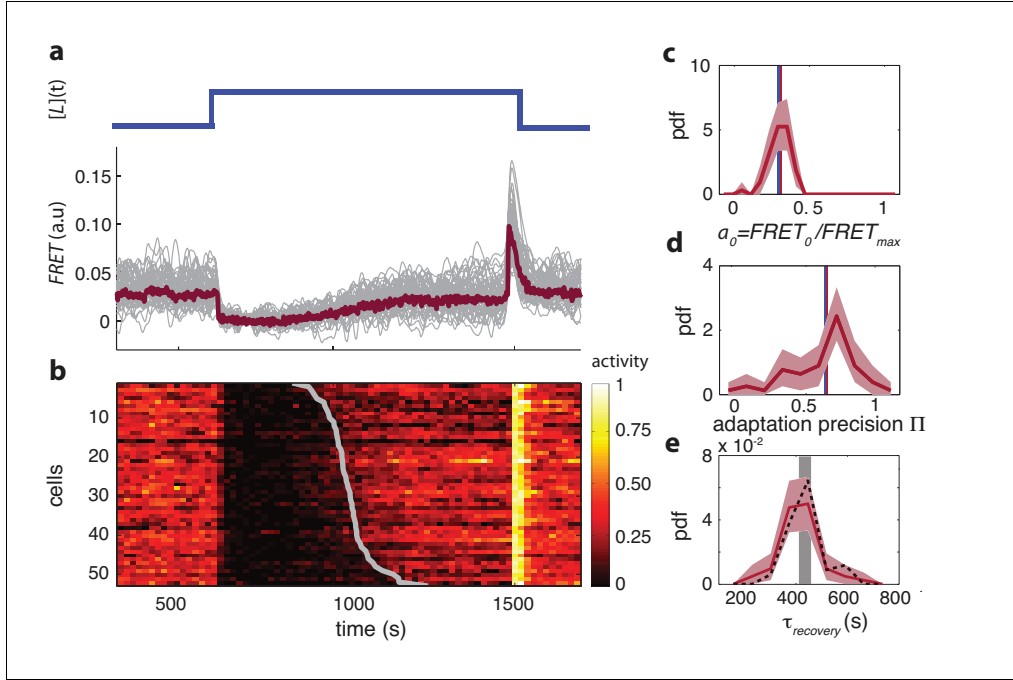

**Figure 1.** Single-cell FRET over extended times reveals cell-to-cell variability in signaling response. (a) Step-response experiment on wildtype cells (CheRB+; VS115). (Top) The ligand time series $[L](t)$ indicates the applied temporal protocol for addition and removal of 500 µM MeAsp. (Bottom) FRET response of 54 cells (grey) with the ensemble-averaged time series (dark red) overlaid from a representative single experiment. Single-cell time series were lowpass filtered with a 14 s moving-average filter. (b) Heatmap representation of the normalized FRET response time series, with each row representing a single cell, and successive columns representing the 10 s time bins in which the color-indicated activity was computed from the FRET time series. Activity was computed by normalizing FRET to the total response amplitude (Max-Min for each time series). Rows are sorted by the corresponding cell's recovery time (grey curve), defined as the time at which the activity recovered to 50% of the activity level after adaptation (see panel e). Single-cell FRET assay schematic and image processing pipeline are shown in *Figure 1—figure supplement 1*. (c) Steady-state activity $a_0$ of the cells shown in panels (a–b). Also shown are the mean steady-state activity (red vertical line) and the steady-state activity of the population averaged time series (blue vertical line). (d) Adaptation precision $\Pi$ obtained from the FRET data. An adaptation precision of 1 denotes perfect adaptation. Also shown are the mean precision (red vertical line) and the precision of the population averaged time series (blue vertical line). The mean and std of the distribution is $0.79 \pm 0.32$. All colored shaded areas represent 95% confidence intervals obtained through bootstrap resampling. (e) Recovery time of cells defined as time to reach 50% of the post-adaptational activity level (red, 54 cells) or 50% of pre-stimulus activity (black dashed, 44 cells with precision >0.5) and simulated effect of experimental noise for a population with identical recovery times (grey). The latter was obtained from a simulated data set in which 55 time series were generated as described in *Figure 1—figure supplement 3*. The width of the bar is defined by the mean ± std of the simulated distribution. The mean ± std of the distributions for the experimental and simulated data sets are respectively $416 \pm 83$ and $420 \pm 35$ s.

DOI: https://doi.org/10.7554/eLife.27455.003

The following source data and figure supplements are available for figure 1:

**Source data 1.** Source data (.mat) file containing FRET data and analysis.

DOI: https://doi.org/10.7554/eLife.27455.007

**Figure supplement 1.** Single-cell FRET assay schematic and workflow.

DOI: https://doi.org/10.7554/eLife.27455.004

**Figure supplement 2.** The observed diversity in signaling parameters can not be explained by variation in experimental parameters and is reproducible.

DOI: https://doi.org/10.7554/eLife.27455.005

**Figure supplement 3.** Influence of experimental noise on estimating recovery times.

DOI: https://doi.org/10.7554/eLife.27455.006

*Kalinin et al., 2010*), but how the amplification and integration varies across a population has not been characterized. To bridge the gap between collective behavior and its underlying single-cell motility it is essential to determine the variability of these important signaling parameters, as well as the origin of the variability. Also, current estimates of the apparent gain in the response (defined as the fractional change in output divided by fractional change in input) are based on population-averaged measurements which may or may not reflect single-cell cooperativity levels. In population averaged measurements, the largest gain is observed in adaptation-deficient (CheRB-) cells (*Sourjik and Berg, 2004*), in which the receptor population is homogeneous with respect to their adaptational modification state and hence in these cells variability in ligand sensing can be studied separately from variability induced by the adaptation enzymes.

We probed the ligand sensitivity of CheRB- cells (TSS58) at the single-cell level by FRET dose-response measurements in which step stimuli of successively larger amplitudes were applied over time (*Figure 2*). Considerable variability in the response to the attractant L-serine were observed across the population of immobilized cells simultaneously experiencing the same stimulus, with response magnitudes often ranging from virtually zero to full response (*Figure 2a*). The resulting dose-response data could be well described by a Hill curve of the form $[1 + ([L]/K)^H]^{-1}$, where the parameters $(1/K)$ and $H$ are defined as the sensitivity and steepness, respectively, of the response of each cell. The family of dose response curves constructed from this ensemble of fit parameters reveals considerable variability from cell to cell in the shape of the response curve (*Figure 2b*).

What could be the cause of the diversity in ligand response in the absence of adaptation-induced heterogeneity? We reasoned that expression-level variability of the five chemoreceptor species of *E. coli*, which are known to form mixed clusters with cooperative interactions (*Ames et al., 2002*; *Sourjik and Berg, 2004*), could endow isogenic populations with sensory diversity. In line with this idea, CheRB- cells expressing only a single chemoreceptor species (Tsr) demonstrated not only higher cooperativity, but also attenuated variability in the dose-response profile from cell to cell (*Figure 2b–c*), showing that the composition of the receptor population is important not only to tune the average ligand response of a population, but also in generating a wide range of sensory phenotypes within an isogenic population.

It has been shown that expression level of chemoreceptors changes during growth of *E. coli* batch cultures: concomitant with the slowing of growth upon the transition from the exponential phase towards early stationary phase, the relative expression level ratio Tar/Tsr, the two most abundant chemoreceptors, increases from majority Tsr (Tar/Tsr<1) to majority Tar (Tar/Tsr>1) (*Salman and Libchaber, 2007*; *Kalinin et al., 2010*). To probe the consequence of such changes for ligand-sensing diversity, we measured single-cell dose response curves in populations harvested at different cell densities during batch growth (*Figure 2d*). The resulting population-averaged responses show a dependence of dose-response parameters on the optical density (OD) of the culture, shifting from highly sensitive (low $K$) and highly cooperative (high $H$) at low cell densities (OD ≈ 0.3) to less sensitive (high $K$) and less cooperative (low $H$) at increased cell densities (OD ≈ 0.45, and OD ≈ 0.6) (*Figure 2d*, open triangles, and *Figure 2—figure supplement 2*). This trend is also visible at the level of single cells, but we found the responses to be highly variable under each condition (*Figure 2d*, filled points). Remarkably, both $K$ and $H$ varied by over an order of magnitude, far exceeding the uncertainty in parameter estimates due to experimental noise (*Figure 2—figure supplement 3*).

To further test the idea that ligand-response diversity is governed by differences in receptor expression levels, we considered the pattern of covariation between the fitted sensitivity $K$ and cooperativity $H$ in single cells (*Figure 2b*, blue). In contrast to cells expressing Tsr as the only chemoreceptor, in which the variability in $K$ is only ~0.15-20% (*Figure 2c*), single cells expressing a wildtype complement of chemoreceptors demonstrated strong variation in $K$. This variation was negatively correlated with the cooperativity $H$ (*Figure 2d*). Noting that this overall pattern of covariation agrees well with dose response parameters obtained from population-level FRET experiments in which the Tar/Tsr ratio was experimentally manipulated via plasmid-based expression control (*Figure 2d*, red circles; data from (*Sourjik and Berg, 2004*)), we proceeded to quantitatively estimate the diversity in the Tar/Tsr ratio via fits of a multi-species MWC model (*Mello and Tu, 2005*; *Keymer et al., 2006*) to single-cell FRET data (see Materials and methods). The resulting distribution of single-cell Tar/Tsr estimates (*Figure 2e*) was dominated by Tsr in cells harvested early (OD ≈ 0.3) but the relative contribution of Tar increased in cells harvested at later stages of growth (OD ≈ 0.45 and OD ≈ 0.6). Interestingly, in addition to this increase in the mean of the Tar/Tsr distribution during batch growth, which confirms previous reports that found

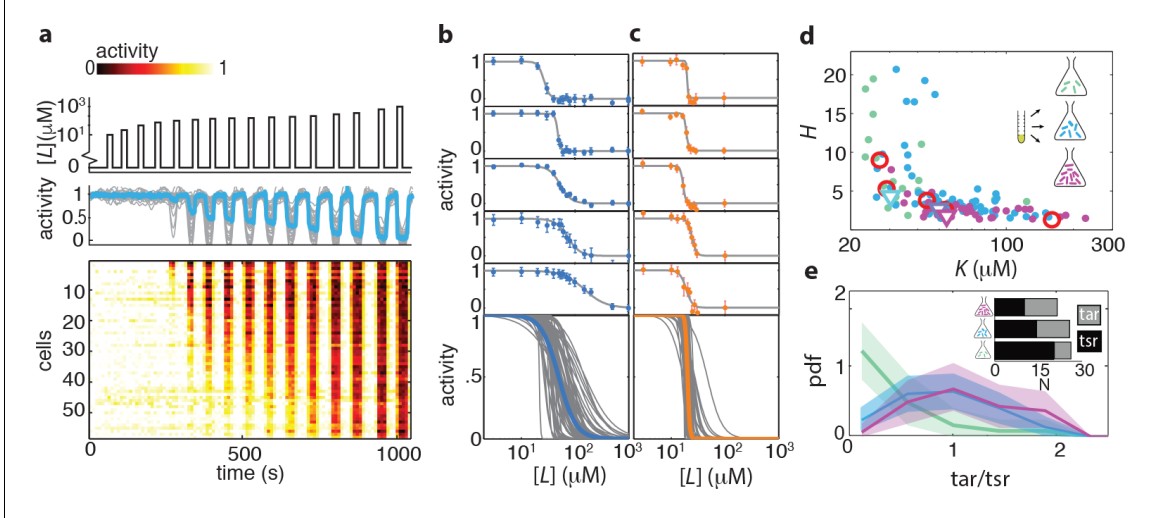

**Figure 2.** Ligand dose-response parameters vary strongly across cells in an isogenic population, even in the absence of adaptation, and depend on receptor-complex composition. (a) Single-cell dose-response experiment on adaptation deficient (CheRB-; TSS58) cells with a wildtype complement of receptors. (Top) Temporal protocol of stimulation $[L](t)$ by the attractant L-serine. (Middle) The ensemble-averaged FRET response of the population (blue) and single cells (gray) in signaling activity of 59 cells from a single experiment, normalized to the full-scale FRET response amplitude. (Bottom) Heatmap representation of the single-cell FRET timeseries, with the rows sorted by the sensitivity $K$ of the corresponding cell obtained from Hill-curve fits. (b) Ensemble of Hill-curve fits (gray) to single-cell dose-response data from a single experiment on CheRB- cells with a wildtype complement of receptors (TSS58). Fits for five example cells from the ensemble are shown above together with data points (error bars: ±2 s.e.m. over 19 frames). The blue curve overlaid on the ensemble was obtained by applying the same analysis to the population-averaged time series shown in panel (a), yielding fit values $K = 50 \pm 3$ μM and $H = 2.7 \pm 0.5$. (c) As in panel (b), but with CheRB- cells expressing only the serine receptor Tsr (UU2567/pPA114). The orange curve was obtained from fits to the population average, yielding $K = 20.0 \pm 0.3$ μM and $H = 22 \pm 8$. (d) Cells from a single overnight culture were inoculated into three flasks harvested at different times during batch-culture growth to sample the state of the population at three points along the growth curve: at $OD_{600} = 0.31$ (green), 0.45 (blue) and 0.59 (purple). Fits to the population-averaged time series are shown in *Figure 2—figure supplement 2*. Shown are Hill-curve sensitivity ($1/K$) and cooperativity $H$ obtained from fits to the single-cell dose-response data, at different harvesting OD's (filled dots) together with the fit values for the population-averaged dose-response data (triangles). Also shown are population-FRET results from (*Sourjik and Berg, 2004*) in which the average Tar and Tsr levels were tuned using inducible promotors (red circles). Shown are 25 out of 28 cells harvested at OD = 0.31, 59 out of 64 cells at OD = 0.45, 34 out of 40 cells at OD = 0.59. The excluded cells had fits with a mean squared error higher then 0.05. The influence of experimental noise on the fit parameters is shown in *Figure 2—figure supplement 3*. (e) Histograms of Tar/Tsr ratio obtained by fitting the multi-species MWC model from (*Mello and Tu, 2005*) to single-cell FRET time series. The mean Tar/Tsr ratios (low to high OD) are 0.4, 0.9, and 1.2 with coefficients of variance of respectively 1.1, 0.5, and 0.4. Inset: average cluster size (MWC-model parameter $N$) of Tar (grey) and Tsr (black) at different harvesting OD's obtained from the fit results in panel d.

DOI: https://doi.org/10.7554/eLife.27455.008

The following source data and figure supplements are available for figure 2:

**Source data 1.** Source data (.mat) file containing FRET data and analysis.
DOI: https://doi.org/10.7554/eLife.27455.012
**Figure supplement 1.** Dose response curve parameters uncertainty estimation and reproducibility.
DOI: https://doi.org/10.7554/eLife.27455.009
**Figure supplement 2.** Dose response curves from population averaged time series at different harvesting OD's.
DOI: https://doi.org/10.7554/eLife.27455.010
**Figure supplement 3.** Influence of experimental noise on fit parameters $K$ and $H$, for Hill curve fits to single-cell dose-response data.
DOI: https://doi.org/10.7554/eLife.27455.011

increased Tar/Tsr ratios at the population level (*Salman and Libchaber, 2007*; *Kalinin et al., 2010*), we find that the breadth of the distribution also increases at later stages of growth. Thus, modulation of receptor expression during growth provides a means of tuning not only response sensitivity and cooperativity, but also single-cell diversity in the response of cell populations experiencing identical changes in their common environment.

The large variability in the Tar/Tsr ratio ($CV \approx 0.5$ at OD=0.45) is somewhat surprising given that the mean expression level of both receptors are known to be high and of order $10^3$-$10^4$ copies per cell (*Li and Hazelbauer, 2004*). At such high expression, intrinsic noise in expression levels (i.e. due to the

production and degradation process of proteins, expected to scale as the square root of the mean) could be as low as a few percent of the mean, and gene-expression fluctuations are expected to be dominated by extrinsic noise components (i.e. those affecting regulation of gene expression, which do not scale with the mean). Quantitative measurements of gene expression reported in previous studies indicate a high degree of covariation among the expression level of chemotaxis genes, both at the population level under changes in growth conditions (*Li and Hazelbauer, 2004*) and at the single-cell level across isogenic cells sampled from the same growth culture (*Kollmann et al., 2005*). Correlated expression-level variation is also expected given the architecture of the flagellar regulon, in which all chemotaxis genes are under the control of a common master regulator (*Chilcott and Hughes, 2000*). These results indicate that the extrinsic (correlated) component of variation is greater than the intrinsic (uncorrelated) variability. Interestingly, however, a recent study (*Yoney and Salman, 2015*) found using single-cell flow-cytometry a high degree of variability in the ratio of Tar/Tsr promotor activities ($CV \approx 0.45$ at OD=0.51) comparable to the range of ratios extracted from our analysis of dose response data. Given that cell-to-cell variability in the Tar/Tsr ratio is much greater than achievable lower bounds of gene-expression noise in bacteria, it would be interesting to investigate the mechanistic sources of this variability, such as operon organization, promotor stochasticity, and translation-level regulatory structures (*Frankel et al., 2014*).

Variability in receptor expression could also explain the distribution of adaptation precision we observed in wildtype cells (*Figure 1d*). In a previous population-level study, it has been shown that adaptation precision depends strongly on the expression-level ratio between the multiple chemoreceptor species, with the highest adaptation precision being achieved when the ligand-binding receptor is a minority within the total receptor population (*Neumann et al., 2014*). Thus, the substantial heterogeneity in adaptation precision we observed ($CV$=0.40) upon a saturating MeAsp stimulus is consistent with strong variability in the Tar/Tsr ratio.

## CheB phosphorylation feedback attenuates cell-to-cell variability

While bacteria can exploit molecular noise for beneficial diversification, variability can also limit reliable information transfer and degrade sensory performance. In the framework of *E. coli*'s run-and-tumble navigation strategy, chemotactic response to gradients requires that cells maintain a finite tumble bias, the fraction of time a bacterium spends tumbling, and avoids extreme values zero and one. The latter cases would correspond to unresponsive phenotypes that fail to switch between run and tumble states in response to the environmental inputs. One important mechanism that ensures responsiveness to stimuli over a broad range of input levels is sensory adaptation mediated by the methyltransferase/methylesterase pair CheR/CheB. These receptor-modifying enzymes provide negative feedback through the dependence of their catalytic activity on the receptor's signaling state: the rate of methylation (demethylation) by CheR (CheB) is a decreasing (increasing) function of receptor-kinase activity (*Borczuk et al., 1986*; *Amin and Hazelbauer, 2010*). This dependence of enzyme activity on the substrate conformation provides negative integral feedback that ensures precise adaptation (*Barkai and Leibler, 1997*) toward the pre-stimulus steady-state activity $a_0$.

Interestingly, one of the two adaptation enzymes, CheB, can be phosphorylated by CheA, the kinase whose activity CheB controls through its catalytic (demethylation) activity on receptors. Effectively, this adds an additional negative feedback loop to the network, but the role of this phosphorylation-dependent feedback has remained elusive since it has been shown to be dispensable for precise adaptation (*Alon et al., 1999*). Through theoretical analysis, it has been conjectured that this secondary feedback loop might play a role in attenuating effects of gene-expression noise (*Kollmann et al., 2005*), but experimental verification has been lacking. We therefore sought to investigate the influence of perturbations to this network topology on the variability of chemotactic signaling activity.

CheB consists of two domains connected by a flexible linker (*Figure 3a*). A regulatory domain, with structural similarity to CheY, can be phosphorylated at residue Asp[56](*Djordjevic et al., 1998*; *Stewart et al., 1990*). A catalytic domain mediates binding to specific residues on chemoreceptor cytoplasmic domains and removes a methyl group added by the counterbalancing activity of CheR. Phosphorylation induces a conformational change and activates CheB (CheB*) (*Djordjevic et al., 1998*; *Lupas and Stock, 1989*). Several mutants of CheB lack phosphorylation feedback while retaining catalytic activity. Here, we focus on two specific mutants: CheB[D56E], which bears a point mutation at the phosphorylation site, and CheB$_c$, which expresses only the catalytic domain of CheB

(*Stewart et al., 1990*; *Alon et al., 1999*). Cells expressing these mutants have an altered network topology (*Figure 3b*) which lacks CheB phosphorylation feedback.

To study the influence of network topology on cell-to-cell variability, we expressed different forms of CheB (CheB$^{\mathrm{WT}}$, CheB$^{\mathrm{D56E}}$, CheB$_c$) from an inducible promoter in a $\Delta cheB$ strain and measured the response to a saturating amount of attractant (500 µM MeAsp). The expression levels of each mutant are tuned such that they approximate the wildtype steady state activity level. The response of CheB$^{\mathrm{WT}}$ was qualitatively very similar to cells in which CheB is expressed from its native chromosomal position (compare *Figure 3—figure supplement 1a* and *Figure 1a*) despite the fact that plasmid expression breaks the translational coupling with CheR (*Løvdok et al., 2009*). By contrast, cells expressing either of the two CheB mutants defective in phosphorylation demonstrated increased cell-to-cell variability in the steady-state activity compared to cells expressing CheB$^{\mathrm{WT}}$. The increased variability of the CheB phosphorylation-deficient mutants (CheB$^{\mathrm{D56E}}$ and CheB$_c$) was manifested not only in a higher coefficient of variation in $a_0$ (1.07 and 1.10, respectively, and WT 0.7), but also a qualitatively different shape of the distribution of $a_0$ across the population (*Figure 3c*). Whereas the distribution demonstrated a single peak in CheB$^{\mathrm{WT}}$ cells with phosphorylation feedback, the distribution for the phosphorylation-feedback mutants demonstrated a bimodal shape with peaks close to the extreme values $a_0=\{0,1\}$.

We tested whether these strong differences in cell-to-cell variability might be the result of gene expression noise, by comparing expression-level distributions of the CheB mutants. We constructed fluorescent fusions of each *cheB* allele to the yellow fluorescent protein mVenus and quantified the distribution of single-cell fluorescence levels under the same induction conditions as in the FRET experiments (*Figure 3—figure supplement 1*). The ratio between the measured expression-levels (CheBc:WT:D56E $\approx$ 0.7:1:2.5) was compatible with expectations from the hierarchy of reported in vitro catalytic rates of CheB ($k_b^{\mathrm{D56E}}<k_b^{WT}<k_b^c$) (*Anand and Stock, 2002*; *Simms et al., 1985*; *Stewart, 1993*), and expression-level variability was very similar between the three strains (*CV*'s of 0.87,0.90 and 0.82; we note that these rather high *CV* values likely include contributions from plasmid copy number variability). These findings suggest that the differences in cell-to-cell variability observed in FRET are not due to differences between the expression-level distributions of the three *cheB* alleles, but rather to the differences they impose on the signaling network topology.

What feature of the signaling network could generate such broad (and even bimodal) distributions of $a_0$? A general paradigm for models of adaptation that exhibit precise adaptation is activity-dependent (integral) feedback (*Barkai and Leibler, 1997*; *Yi et al., 2000*), which in bacterial chemotaxis can be implemented by the activity of the feedback enzymes CheR and CheB being dependent of the conformational state (i.e. activity) of their substrate chemoreceptors. This results in a steady-state activity $a_0$ that only depends on the [R]/[B] expression-level ratio and not on their absolute abundance. We can view this mapping as a transfer function $f$ between the ratio [R]/[B] and the steady-state activity,

$$a_0 = f([\mathrm{R}]/[\mathrm{B}])$$

Depending on the function $f$, the input variance $P_{\mathrm{RB}}([\mathrm{R}]/[\mathrm{B}])$ may lead to high or low variance in the distribution $P(a_0)$. This is because the manner in which the transfer function $f$ filters the [R]/[B] distribution,

$$P(a_0) = \frac{P_{RB}(f^{-1}(a_0))}{|f'(f^{-1}(a_0))|}.$$

Hence a steep function $f$ can impose bimodality in the methylation level, and thereby also in the activity of steady-state CheA activity, $a_0$, even at quite modest input variances for distributions of the ratio [R]/[B].

Thus, even if expression-level noise for both CheR and CheB are modest, a sensitive transfer function $f$ can effectively amplify the variation in [R]/[B], and if the distribution of the latter ratio, $P_{\mathrm{RB}}([\mathrm{R}]/[\mathrm{B}])$ extends below and above the narrow region over which $f$ is steep, the decreased slope of $f$ (i.e. lower $f'([\mathrm{R}]/[\mathrm{B}])$ in those flanking regions will tend to increase the weight on both sides of the broad $P(a_0)$ distribution to produce a bimodal profile. On the other hand, if the network topology effectively reduces the steepness of $f$, the resulting $P(a_0)$ will have a reduced variance for the same input $P_{\mathrm{RB}}([\mathrm{R}]/[\mathrm{B}])$ (*Figure 3d*). Our results suggest that $f$ is much steeper in the absence of phosphorylation feedback than in its presence.

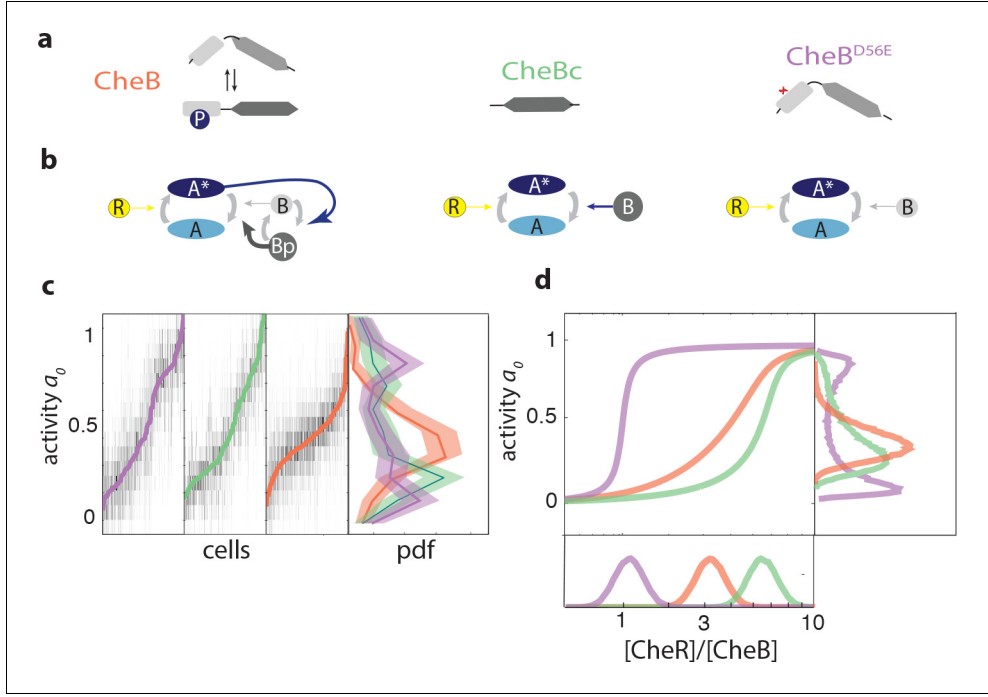

**Figure 3.** CheB phosphorylation feedback attenuates variability in steady-state kinase activity. (**a**) Schematic depiction of CheB activation by phosphorylation. (Top) CheB consists of two domains connected by a flexible linker. The aspartate at residue 56 within the N-terminal receiver domain can be phosphorylated. (Middle) CheBc lacks the receiver domain with the phosphorylation site. (Bottom) CheB-D56E carries a point mutation at the phosphorylation site. (**b**) Effective network topology of cells expressing WT CheB (top), CheBc (middle) and CheB-D56E (bottom). All three topologies are capable of precise adaptation due to activity-dependent feedback (**Barkai and Leibler, 1997**). (**c**) Heatmap representation of histograms of the activity $a(t)$ about the unstimulated steady-state of single cells, from FRET experiments of the type shown in **Figure 3—figure supplement 1**. Each column represents a single cell, sorted by the steady-state activity $a_0$ (colored curves) for each CheB mutant expressed in a *cheB* background (VS124, colors as in panel (**a**)). (right) Normalized histograms (probability density function, pdf) of $a_0$ for each CheB mutant. Histograms contain results for cells with a signal-to-noise ratio greater than one from at least three independent FRET experiments, corresponding to 231 out of 280 cells (WT), 169 out of 210 cells (CheBc) and 156 out of 246 cells (D56E). Shaded regions represent bootstrapped 95% confidence intervals. We verified that the bimodality was not due to clipping from FRET-pair saturation, by mapping the dependence of FRET on donor/acceptor expression (**Figure 3—figure supplement 2**). (**d**) A simple kinetic model of the chemotaxis network illustrates the crucial role of CheB phosphorylation feedback in circumventing detrimental bimodality in $a_0$. Due to sublinear enzyme kinetics in the adaptation system, the transfer function $a_0 = f([R]/[B])$ mapping the P([R]/[B]) expression ratio to steady-state network output $a_0$ can be highly nonlinear (main panel). The shape of this transfer function determines the distribution $P(a_0)$ of steady-state activity (right panel) by transforming the distribution $P([R]/[B])$ of adaptation-enzyme expression ratios (bottom panel). Three variations of the model are shown, corresponding to WT (orange, with phosphorylation feedback), CheB[D56E] (purple, no phosphorylation feedback and low catalytic rate), and CheBc (green, no phosphorylation feedback, high catalytic rate).

DOI: https://doi.org/10.7554/eLife.27455.013

The following source data and figure supplements are available for figure 3:

**Source data 1.** Source data (.mat) file containing FRET data and analysis.
DOI: https://doi.org/10.7554/eLife.27455.019
**Figure supplement 1.** Example FRET time series and CheB localization.
DOI: https://doi.org/10.7554/eLife.27455.014
**Figure supplement 2.** Relation between maximum FRET response and expression levels of donor and acceptor fluorophores.
DOI: https://doi.org/10.7554/eLife.27455.015
**Figure supplement 3.** Linear and supralinear models of CheB feedback cannot explain bimodality in $a_0$.
DOI: https://doi.org/10.7554/eLife.27455.016

*Figure 3 continued on next page*

*Figure 3 continued*
**Figure supplement 4.** Phosphorylation feedback is not a necessary condition for fast removal adaptation dynamics.
DOI: https://doi.org/10.7554/eLife.27455.017
**Figure supplement 5.** Phosphorylation defective mutants show impaired chemotaxis on soft agar.
DOI: https://doi.org/10.7554/eLife.27455.018

We find that models with linear or supra-linear dependence of the methylation rate on activity generate a function $f$ that is very shallow (*Figure 3—figure supplement 3*), making them unsuitable for explaining the observed bimodal behavior. However, if we assume CheR and CheB follow Michaelis-Menten kinetics in which the dependence of the methylation rates on receptor activity is sub-linear, the dependence of $f$ on [R]/[B] can become very steep. It has been conjectured (*Barkai and Leibler, 1997*; *Emonet and Cluzel, 2008*) that *in vivo* the enzymes CheR and CheB operate at or near saturation, an idea supported by population-level FRET measurements of adaptation kinetics (*Shimizu et al., 2010*). An important consequence of enzyme saturation in such reversible modification cycles is that the steady-state activity of the substrate can become highly sensitive to the expression level ratio of the two enzymes, a phenomenon known as zero-order ultrasensitivity ((*Goldbeter and Koshland, 1981*); see Materials and methods). Within the chemotaxis system, saturation of both CheR and CheB can thus render the receptor modification level, and in turn, the CheA activity $a_0$, ultrasensitive to the [R]/[B] concentration ratio (*Emonet and Cluzel, 2008*).

Could the known biochemical differences between the three forms of CheB (CheB$^{WT}$, CheB$^{D56E}$, CheB$_c$) explain the contrasting patterns of $a_0$ variability observed in our single-cell FRET experiments? In the absence of any feedback, the steepness of $f'$([R]/[B]) is solely determined by the low Michaelis-Menten constants $K_{B,R}$, which corresponds to saturated kinetics of the enzymatic activity of CheRB and hence ultransensitivity of the steady-state substrate activity. The expression ratio of CheR/CheB which determines the crossover point ($a_0$=0.5) is set by the ratio of catalytic rates of CheR and CheB ($k_{r,b}$). Hence the phosphorylation deficient mutants CheB$^{D56E}$ and CheB$_c$ both have steep curves but are shifted along the R/B axis due to very different catalytic rates. However, in the case of phosphorylation feedback, CheB$^{WT}$, the same enzyme can be in two states, each with equal $K_{r,b}$ but one low and one high $k_r$. Whether CheB is in the one state or the other is determined by the activity-dependent phosphorylation feedback. As a result, the curve of CheB$^{WT}$ is activity dependent ($f(a,[R]/[B])$) and changes with activity by shifting between the two curves corresponding to the extremes of all phosphorylated or all unphosphorylated. Effectively, this makes the resulting curve $f$ less steep (*Emonet and Cluzel, 2008*). The mean of the distributions $P_{RB}$ are tuned such to get the same mean activity level ($\langle a_0 \rangle$), but the same variance in $P_{RB}$ leads to very wide $P(a_0)$ distributions in absence of phosphorylation, while phosphorylation feedback ensures a much smaller, single-peaked distribution.

It has also been conjectured that the CheB phosphorylation feedback is responsible for the highly nonlinear kinetics of recovery from repellent (or attractant removal) responses (*Shimizu et al., 2010*; *Clausznitzer et al., 2010*). Indeed, in cells expressing CheB$_c$, the kinetics of recovery from the response to removal of 500 µM MeAsp after adaptation appeared qualitatively different from that in cells expressing wildtype CheB, lacking the characteristic rapid recovery and instead appearing more symmetric with the CheR-mediated recovery upon addition of a saturating dose of attractant (*Figure 3—figure supplement 4*). By contrast, CheB$^{D56E}$ was found to still possesses a fast component, despite being defective in phosphorylation, albeit also with somewhat slower kinetics than wt. In summary, the clearest difference between wildtype and phosphorylation-defective CheB mutants is found in the variability of the steady-state signal output (i.e. kinase activity).

The bimodal distribution in kinase activity we observed in the phosphorylation-deficient CheB mutants implies that a large fraction of cells have a CheY-P concentration far below or far above the motor's response threshold and hence will impair chemotactic responses to environmental gradients. Consistent with this idea, in motility-plate experiments (Supplementary *Figure 3—figure supplement 5*) we found that chemotactic migration on soft-agar plates was severely compromised for both CheB$^{D56E}$ and CheB$_c$ compared to CheB$^{WT}$, indicating that the phosphorylation feedback is important for efficient collective motility.

## Protein-signaling noise generates large temporal fluctuations in network output

The slow kinetics of the adaptation enzymes CheR and CheB have been hypothesized to play a role not only in determining the steady-state kinase activity $a_0$, but also in generating temporal fluctuations of the intracellular signal (*Korobkova et al., 2004*; *Emonet and Cluzel, 2008*; *Park et al., 2010*; *Celani and Vergassola, 2012*). We found substantial differences between wildtype (CheRB+) and adaptation-deficient (CheRB-) cells in the variability of their FRET signals across time (*Figure 4*). The effect is clearly visible upon comparing long (~1 hr) FRET time series obtained from cells of these two genotypes (*Figure 4a*). The FRET signal in wildtype cells demonstrated transient excursions from the mean level that were far greater in amplitude than those in CheRB- cells. To distinguish between variability across cells in a population (which we discuss in terms of coefficients of variation, *CV*) and that over time within a single cell, we denote the temporal noise amplitude as $\eta \equiv \sigma_a/a_0$. This amplitude was quantified by computing the variance of each single-cell time series, low-pass filtered with a moving average filter of 10 s, and shows that the fluctuation amplitudes are much larger in wildtype cells compared to adaptation-deficient cells ($\langle\eta\rangle$ = 0.44 and 0.09 respectively, *Figure 4b*). Importantly, these experiments were carried out under conditions in which no protein synthesis can occur due to auxotrophic limiation (see Materials and methods), thus ruling out gene-expression processes as the source of these fluctuations.

Power spectral density (PSD) estimates computed from such time series confirm a nearly flat noise spectrum for CheRB- cells, whereas CheRB+ cells demonstrated elevated noise at low frequencies (*Figure 4c*). The amplitude of these low-frequency noise components do clearly vary from cell to cell, as can be gleaned in the diversity of single-cell power spectra. To quantify this protein-level noise due to CheR/CheB activity, we describe the fluctuating signal as an Ornstein-Uhlenbeck (O-U) process of the single variable $a$, with relaxation timescale $\tau$ and diffusion constant $c$, which can be interpreted as a linear-noise approximation (*Van Kampen, 1981*; *Elf and Ehrenberg, 2004*) to the multivariate stochastic kinetics of the underlying chemical network controlling the mean kinase activity $a$(*Tu and Grinstein, 2005*; *Emonet and Cluzel, 2008*):

$$\frac{da}{dt} = -\frac{1}{\tau_m}a(t) + \sqrt{c}\Gamma(t) \tag{1}$$

where $\Gamma(t)$ is a Gaussian white noise process. The parameters $\tau_m\tau_m$ and $c$ for each cell are readily extracted via the power-spectrum solution of the O-U process:

$$S_a(\omega) = \frac{2c\tau^2}{1 + (2\pi\omega\tau_m)^2} + E \tag{2}$$

where we have added to the standard Lorentzian solution (*Gillespie, 1996*) a white-noise term $E$ that may vary from cell to cell to account for experimental shot noise in the photon-limited FRET signal. Single-cell PSD data were well fit by *Equation 2* (*Figure 4d*), and the average of extracted single-cell fluctuation timescales ($\langle\tau_m\rangle = 12.6$s) (*Figure 4e*) are in good agreement with previously reported correlation times of flagellar motor switching (*Park et al., 2010*; *Korobkova et al., 2004*), as well as the kinetics of CheRB-mediated changes in receptor modification from *in vivo* measurements using radioactively labeled methyl groups (*Lupas and Stock, 1989*; *Terwilliger et al., 1986*). The variance of the fluctuations obtained from the fits of the PSD, $\sigma_a = \sqrt{c\tau_m/2}$ yielded very similar noise amplitudes $\eta_{OU} \equiv \sigma_{a,OU}/a_0$ as calculated from the time series ($\langle\eta_{OU}\rangle = 0.42$, *Figure 4—figure supplement 3*). We note that these noise levels are larger than expected - in a considerable fraction of cells, the standard deviation of fluctuations is comparable to the mean level of activity, and the steady-state fluctuations span the full range of kinase activity (see e.g. that represented by the red curve in *Figure 4a*). Previous studies had predicted a value of ~10-20%, based either on reported fluctuation amplitudes of motor switching (*Korobkova et al., 2004*; *Tu and Grinstein, 2005*) or biochemical parameters of the intracellular signaling network (*Emonet and Cluzel, 2008*; *Shimizu et al., 2010*). The noise amplitudes are also highly variable (*CV*=0.55, $\sigma_\eta$=0.24) from cell to cell.

In summary, we confirmed the presence of strong temporal fluctuations in single-cell chemotaxis signaling attributable to the stochastic kinetics of the adaptation enzymes CheR/CheB, and further

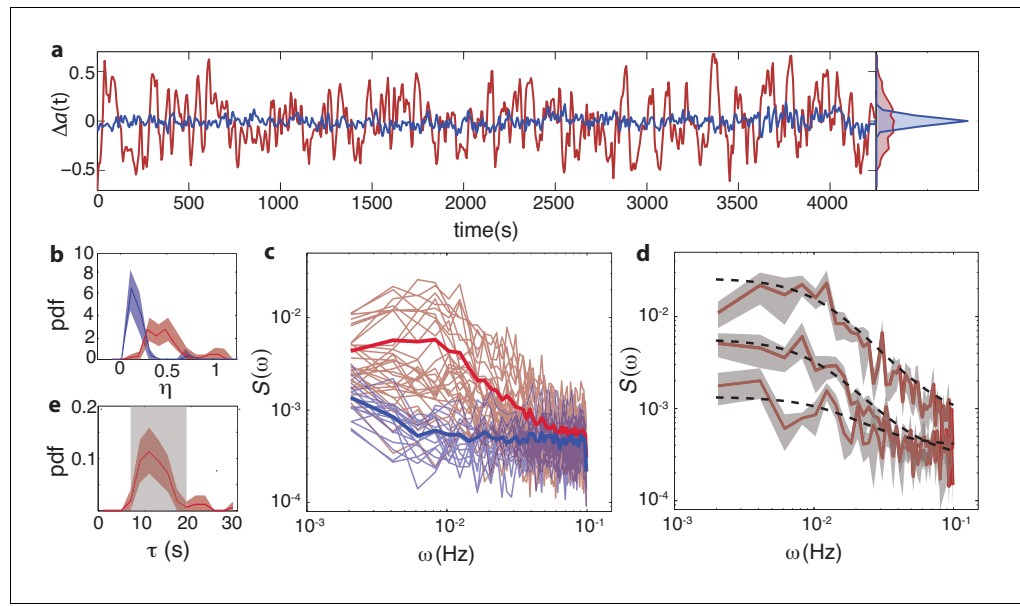

**Figure 4.** Temporal fluctuations in WT cells due to stochastic activity of adaptation enzymes CheR/CheB. (a) Representative single-cell FRET time series of steady-state fluctuations $\Delta a(t) = a(t) - a_0$ in WT cells (VS115, red), together with analogous data from CheRB- cells (TSS58,blue) for comparison (low-pass filtered with a 10 s moving average filter). (b) Histogram of fluctuation amplitude $\eta$ ($\equiv \sigma_a/a_0$) for both WT (89 cells, red, from three independent experiments) and CheRB- (33 cells, blue, from two independent experiments), extracted from calculating the standard deviation of a low-pass filtered FRET time series over a 10 s window divided by the mean FRET level of a single cell. Shaded areas represent 95% confidence intervals obtained from bootstrap resampling. (c) Power spectral density (PSD) computed from single-cell FRET time series of 31 WT cells (red, from single experiment) and 17 CheRB- cells (blue, from single experiment), each from a single experiment. Thin curves in the lighter shade of each color represent single-cell spectra, and their ensemble average is shown as thick curves in a darker shade. The increased power at low-frequencies in WT cells was lost when PSD was computed after ensemble-averaging the time series *Figure 4—figure supplement 1*, indicating that these slow fluctuations are uncorrelated across cells. (d) Representative single-cell PSDs and fits by an Ornstein-Uhlenbeck (O–U) process. Shown are O-U fits (Lorentzian with constant noise floor; dashed curves) to three single-cell PSDs (solid curves). Shaded areas represent standard errors of the mean for PSDs computed from nine non-overlapping segments of each single-cell time series. Fits to all cells from the same experiment are shown in (*Figure 4—figure supplement 2*). Noise amplitudes computed from the O-U fit parameters (*Figure 4—figure supplement 3*) demonstrate excellent agreement with those computed directly from the time series (panel b). (e) Histogram of fluctuation timescales $\tau$ extracted from O-U fits to single-cell PSDs (red, 75 out of 89 cells). Cells without a clear noise plateau at low frequencies were excluded from the analysis (*Figure 4—figure supplement 3*). Red shaded region represents 95% confidence intervals obtained from bootstrap resampling. The gray shaded region indicates the variability (mean±std) that can be explained by experimental noise and a finite time window, obtained from simulated O-U time series (see Materials and methods).

DOI: https://doi.org/10.7554/eLife.27455.020

The following source data and figure supplements are available for figure 4:

**Source data 1.** Source data (.mat) file containing FRET data and analysis.
DOI: https://doi.org/10.7554/eLife.27455.024

**Figure supplement 1.** PSD estimates from population-averaged time series.
DOI: https://doi.org/10.7554/eLife.27455.021

**Figure supplement 2.** Fits of OU process to PSD estimates from single-cell FRET time series from a single experiment.
DOI: https://doi.org/10.7554/eLife.27455.022

**Figure supplement 3.** Comparison between noise amplitudes obtained from time series and power spectra and reproducibility of noise characteristics between different experiments.
DOI: https://doi.org/10.7554/eLife.27455.023

found that the amplitude of these fluctuations vary considerably across cells in an isogenic population.

## Receptor-kinase fluctuations in the absence of adaptation reveal two-level switching

The fluctuation amplitude $\eta$ in CheRB+ cells (*Figure 4b*) is much greater than previous estimates from pathway-based models that considered sublinear kinetics in the enzymatic activities of CheR and CheB (*Emonet and Cluzel, 2008*) and receptor cooperativity (*Shimizu et al., 2010*) as possible mechanisms that amplify noise originating in the stochastic kinetics of receptor methylation/demethylation. A possible explanation for this discrepency is the presence of one or more additional noise source(s) independent of methylation/demethylation dynamics. Although we found that the noise amplitude $\eta$ was much lower than wildtype in unstimulated CheRB- cells (*Figure 4*), it is possible that the strong activity bias of these cells in the absence of chemoeffectors ($a_0 \approx 1$) masks noise contributions that would be observable if receptors were tuned to the more responsive regime of intermediate activity (e.g. as in wt cells, where $a_0 \approx 1/3$). We reasoned that in CheRB- cells, tuning the activity to an intermediate level by adding and sustaining a sub-saturating dose of attractant could reveal additional noise sources. Hence we measured the temporal variability of CheRB- cells during prolonged stimulation with 50 µM L-serine, which elicits a half-maximal population-level response (*Figure 5b*). Although no large fluctuations were be observed in the population-averaged time series (*Figure 5b*), averaging the power spectra computed from all single-cell time series revealed a somewhat elevated noise level at low frequencies, compared to the case without ligand (*Figure 5a*), indicating the possibility of a noise source independent of receptor methylation.

To further test whether and how these methylation-independent fluctuations are affected by the composition of the chemoreceptor arrays, we also measured the response of CheRB- cells expressing Tsr as the sole chemoreceptor during a sustained stimulus of magnitude close to the population-level $K$ (*Figure 5c*). Surprisingly, the averaged single-cell power spectra (*Figure 5a*) indicated the presence of very large fluctuations, even surpassing the fluctuation magnitude in CheRB+ cells. The time series of single-cell responses demonstrated strong deviations from the population average (*Figure 5d and  - Video Supplement*). Whereas all cells responded identically to the saturating dose of attractant, the behavior during the sub-saturating step was highly diverse. Some cells (11/141) showed no apparent response in kinase activity, whereas in others (32/141) complete inhibition was observed (*Figure 5d*, yellow curves). The majority of cells (98/141), however, had an intermediate level of activity when averaged over time, but demonstrated strong temporal fluctuations, often with magnitudes exceeding those observed in wildtype cells.

We further noted that within this subset of cells with large temporal fluctuations, a large fraction (54/98) demonstrated fluctuations that resemble rapid step-like transitions between discrete levels of relatively stable activity that could be identified as peaks in the distribution of activity values across time (*Figure 5d*, marginal histograms). Among these 'stepper' cells, the majority (37/54) appeared to transition between three or more discrete activity levels (*Figure 5d*, brown curve), whereas the remaining sizable minority of steppers (17/54) demonstrated binary switching between two discrete levels corresponding to the maximum ($a \approx 1$) and minimum ($a \approx 0$) receptor-kinase activity states (*Figure 5d*, red curve). The remaining fraction of cells (44/98) demonstrated fluctuations that were also often large but in which discrete levels could not be unambiguously assigned (*Figure 5d*, black curve). The numbers of cells corresponding to each of the categories described above are summarized in *Figure 5e*.

The observation of cells that demonstrate spontaneous two-level switching is particularly surprising, given the large number of molecules involved in receptor-kinase signaling. The expression level of each protein component of the chemoreceptor-CheW-CheA signaling complex in our background strain (RP437) and growth medium (TB) has been estimated (by quantitative Western Blots) to be of order $10^4$ copies/cell (*Li and Hazelbauer, 2004*). Considering that the core unit of signaling has a stoichiometric composition of receptor:W:A = 12:2:2 (monomers) (*Li and Hazelbauer, 2011*), the number of core units is likely limited by the number of receptors, leading to an estimate $10^4/12{\sim}10^3$ core units for a typical wildtype cell. This estimate does not apply directly to the experiments of *Figure 5* because receptors are expressed from a plasmid in a strain deleted for all receptors. But the FRET response amplitudes of these cells were similar to those of cells with a wildtype complement

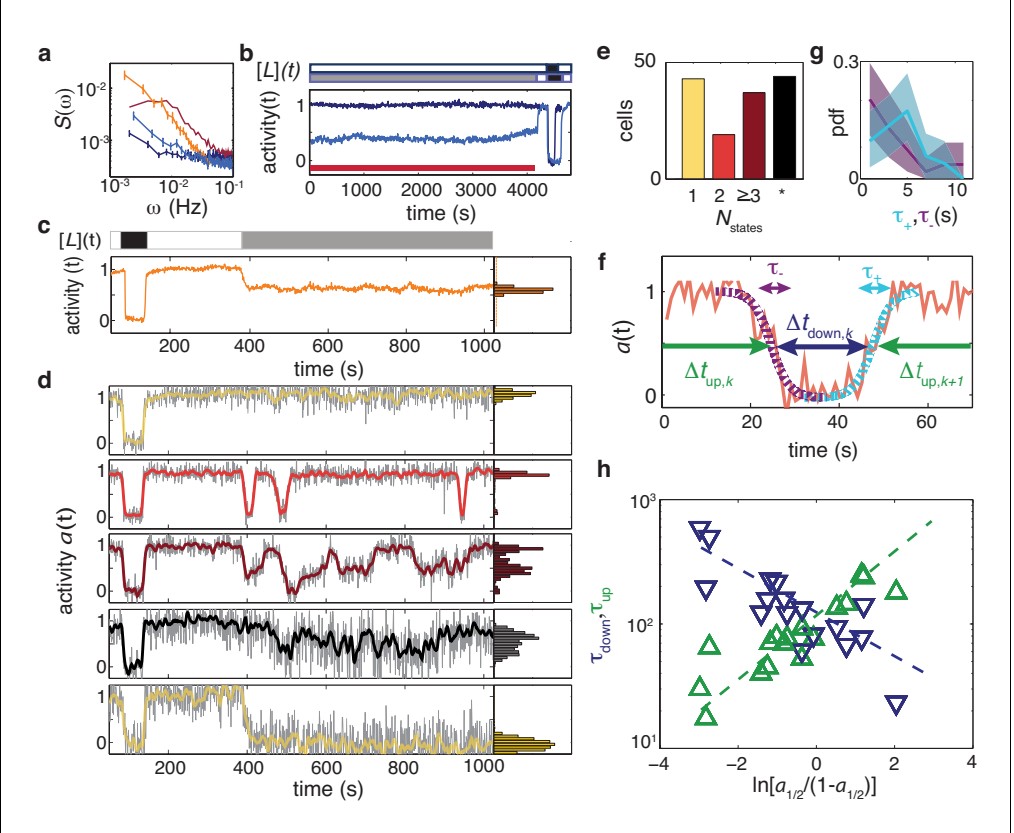

**Figure 5.** Temporal fluctuations in adaptation-deficient cells depend strongly on activity and composition of chemoreceptor population. (a) Power spectral density (PSD) for temporal signal fluctuations during sub-saturating ligand stimulation of 18 cells with wild-type receptor complement (light blue, CheRB-, TSS58) and 58 cells expressing only chemoreceptor Tsr (orange, CheRB- Tsr+, TSS1964/pPA114). Also shown, for comparison, are PSDs from experiments without ligand stimulation for WT cells and CheRB- cells (red and dark blue, respectively; same data as in **Figure 4**). Error bars represent standard error of the mean. We note that the Tsr+ experiment had a larger FRET amplitude scaling factor $\mathrm{FRET_{max}}$ (see Materials and methods) compared to the standard conditions under which the other strains were measured, and to account for this difference, the Tsr+ power spectrum has been scaled by a factor $\rho^2 = 0.17$, where $\rho \equiv \mathrm{FRET_{max,std}}/\mathrm{FRET_{max,Tsr+}}$ to account for this difference. (b) (Top) Stimulus protocol for modulation of the L-serine ligand concentration ($[L](t)$). Cells were incubated either in buffer ($[L]=0$, white) or a subsaturating stimulus ($[L] = 50\ \mu M$, gray) for 1> hr. A saturating stimulus ($[L] = 1\ mM$, black) is applied at the end of the experiment. (Bottom) Population- averaged time series for adaptation-deficient cells with wildtype receptor complement (CheRB-, TSS58) for experiments with (18 cells, light blue) and without (17 cells, dark blue) a sustained 50 $\mu M$ L-serine stimulus during the time interval used to compute the PSDs in panel a (indicated by the red bar). (c) (Top) Stimulus protocol for L-serine concentration ($[L](t)$). At the start of the experiment, a saturating concentration ($[L] = 1\ mM$, black) is applied for a short time. After flushing buffer ($[L]=0$, white), an intermediate concentration ($[L] = 25\ \mu M$, gray) is sustained for 10 min. (Bottom) Population-averaged time series of 58 adaptation-deficient cells expressing Tsr as the sole chemoreceptor (RB-Tsr+; TSS1964/pPA114) under the stimulus protocol indicated above. (d) Selected single-cell time series of the population shown in panel (c), each normalized to its activity level before adding the first stimulus. To the unfiltered data (gray) a 7 s moving average filter is applied and superimposed (colored according to categories in panel (e)). All time series and corresponding activity histograms of the same experiment are shown in **Figure 5 - Supplement 1** and **5**. (e) Classification of RB-Tsr+ single cell fluctuation phenotypes by the number of stable activity levels observed during the sustained subsaturating stimulus. Many cells show only one stable activity level (yellow), corresponding to either full-amplitude response ($a \to 0$) or no response ($a \to a_0$). Some cells show two (red) or more (purple) apparently stable states. In other cells, fluctuations appeared chaotic with no discernibly stable state (black). (f) Definitions for analysis of two-state switching dynamics. The transition timescales $\tau_+$ and $\tau_-$ were determined by fits of a symmetric exponential function (see main text) to the upward (cyan) and downward (purple) switching transients, respectively. Residence times $\Delta t_{up,down}$ were defined as the interval between two successive transitions, at 50% activity. (g) Histogram of transition timescales, $\tau_+$ (4.2 ± 2.2 s, 26 events, cyan) and $\tau_-$

*Figure 5 continued on next page*

*Figure 5 continued*

(3.5 ± 3.2, 29 events, purple) from 10 two-state switching cells of a single experiment with 1 Hz acquisition frequency. (h) Mean residence times $\tau_{\mathrm{up}}$ and $\tau_{\mathrm{down}}$ for two-state switching cells as a function of the average activity bias $\ln[a_{1/2}/(1 - a_{1/2})]$. The slopes are $\gamma_{\mathrm{down}} = -0.4$ and $\gamma_{\mathrm{up}} = 0.6$, and the crossover point at $\tau_{\mathrm{up}} = \tau_{\mathrm{down}} = 110 \pm 10$ s defines a characteristic switching timescale. Data of 17 cells from three independent experiments (one at 1 Hz acqusition, two at 0.2 Hz acquisition).
DOI: https://doi.org/10.7554/eLife.27455.025

The following video, source data, and figure supplements are available for figure 5:

**Source data 1.** Source data (.mat) file containing FRET data and analysis.
DOI: https://doi.org/10.7554/eLife.27455.028

**Figure supplement 1.** All single-cell time series from a single representative experiment.
DOI: https://doi.org/10.7554/eLife.27455.026

**Figure supplement 2.** Histograms of activity during attractant stimulation for all cells from a single representative experiment.
DOI: https://doi.org/10.7554/eLife.27455.027

**Figure 5—video 1.** Segmentation video of three cells showing stochastic switching dynamics.
DOI: https://doi.org/10.7554/eLife.27455.029

of receptors, and we thus expect the number of active core units per cell in the experiments of *Figure 5* to be similar to or greater than that in wildtype cells.

We analyzed further the temporal statistics of the discrete transitions in the subset of cells exhibiting two-level switching (*Figure 5g–h*). We first quantified the duration of such transitions by fitting segments of the activity time series over which these switches occured (*Figure 5d*) by a symmetrized exponential decay function (see Materials and methods) to obtain switch durations $\tau_+$ and $\tau_-$ for upward and downward transitions, respectively. The fitted values for $\tau_+$ and $\tau_-$ correspond to the duration over which the activity trajectory traverses a fraction $1 - e^{-1}$ of the transition's full extent, and were found to be similar between switches in both directions: $\langle\tau_+\rangle \pm \sigma_{\tau_+} = 4.2 \pm 2.2$ s and $\langle\tau_-\rangle \pm \sigma_{\tau_-} = 3.5 \pm 3.2$ s (*Figure 5e*). We note that these transition times are significantly greater than, but close to, the data acquisition interval (1 s), and so the shape of the fitted function should be considered a first approximation to the true rise and decay dynamics.

We then considered the duration of time between switching events. We defined $\Delta t_{\mathrm{up},k}$ and $\Delta t_{\mathrm{down},k}$ as the duration of the $k$-th time interval between transitions with high- and low-activities, respectively (*Figure 5d*), and computed the average over all $k$ of $\Delta t_{\mathrm{up/down},k}$ for each individual cell to estimate its residence timescales $\tau_{\mathrm{up/down}}$ for states of high/low activity, respectively. From each cell's set of intervals $\{\Delta t_{\mathrm{up/down},k}\}$ we also computed a parameter $a_{1/2}$, defined as the fraction of time the cell spent in the high activity level, as a measure of its time-averaged activity during the sub-saturating (20M) L-serine stimulus that yielded a population-averaged response $\langle a\rangle \approx 1/2$ (see Materials and methods).

We found that the logarithms of the mean residence times $\tau_{\mathrm{up}}$ and $\tau_{\mathrm{down}}$ scale approximately linearly with $\ln[a_{1/2}/(1 - a_{1/2})]$ (*Figure 5f*). The latter can be considered a free-energy difference $(-\Delta G) = G_{\mathrm{down}} - G_{\mathrm{up}}$ between the inactive and active states of an equilibrium two-state switching process in which the time-averaged activity $a_{1/2}$ is given by the probability of being in the active state, $a_{1/2} = p(\mathrm{active}) = (1 + e^{\Delta G})^{-1}$. The residence time in each state can then be described by an Arrhenius-type relation with characteristic time for barrier crossing $\tau_r$ and the height of the energy barrier dependent on $\Delta G$,

$$\begin{aligned}\tau_{\mathrm{down}} &= \tau_r \exp\left[-\gamma_{\mathrm{down}}\Delta G/k_B T\right] \\ \tau_{\mathrm{up}} &= \tau_r \exp\left[-\gamma_{\mathrm{up}}\Delta G/k_B T\right]\end{aligned} \tag{3}$$

where the (dimensionless) constants $\gamma_{\mathrm{down}}$ and $\gamma_{\mathrm{up}}$ describe how the barrier heights of the down and up states, respectively, depend on the free-energy difference $\Delta G = k_B T \ln[(1 - a_{1/2})/a_{1/2}]$. We find $\gamma_{\mathrm{down}} = -0.4 \pm 0.1$, $\gamma_{\mathrm{up}} = 0.6 \pm 0.1$, and the characteristic timescale $\tau_r$, defined here as equivalent to $\tau_{\mathrm{up}} = \tau_{\mathrm{down}}$ when $\Delta G = 0$ (and hence $a_{1/2} = 0.5$), was found to be $110 \pm 10$ s. The fact that the mean residence times $(\tau_{\mathrm{up}}, \tau_{\mathrm{down}})$ scale exponentially with the apparent free energy difference $(\Delta G)$

indicates that receptor-kinase switching can, to a first approximation, be treated as a barrier-crossing process.

In summary, these data demonstrate the existence of a signaling noise source that is independent of the adaptation enzymes CheR/CheB. The fluctuations they generate can be very strong in cells expressing Tsr as the sole chemoreceptor, leading to two-level switching in a subset of cells. The latter observation suggests that cooperativity among signaling units in homogeneous chemoreceptor arrays can reach extremely high values, with up to $\sim 10^3$ units switching in a cooperative fashion. The temporal statistics of these two-level switches are consistent with a barrier-crossing model in which the residence time of both states depend on the activity bias $\ln[a_{1/2}/(1 - a_{1/2})]$ in a nearly symmetric manner with opposing signs.

## Discussion

The single-cell FRET measurements described here allowed us to quantify variability in a variety of signaling parameters of the bacterial chemotaxis system, both across cells in a population and within individual cells over time. The magnitude of variation was large across a wide variety of signaling parameters, as summarized in *Table 1*. By imaging many (typically ~50) cells simultaneously, we are able to record signaling in individual cells at high throughput, to build up single-cell statistics. Although single-cell experiments have a long history in studies of bacterial chemotaxis (*Berg and Brown, 1972*; *Spudich and Koshland, 1976*; *Block et al., 1982*; *Korobkova et al., 2004*; *Dufour et al., 2016*), nearly all examples to date have relied on measurements of flagellar motor output (in either tethered or swimming cells). A major advantage of the FRET approach is that it provides a direct measurement of intracellular signaling that bypasses the noisy behavior of the flagellar motor (a stochastic two-state switch), thereby enabling accurate and efficient determination of signaling parameters. The anti-parallel response signature of ratiometric FRET provides a good way to discriminate genuine FRET changes from imaging artifacts. As in population-level FRET, single-cell FRET is most easily applied to study large and rapid changes in signaling (e.g. response to step stimuli), but we have shown that with careful correction of drifts in the signal level (primarily due to bleaching, but also including contributions from fluorophore maturation and/or recovery from long-lived dark states), it can be applied effectively to measure more subtle changes in signaling over extended times, including steady-state fluctuations. Care is required in these corrections of long-time fluorescence-intensity drifts because imperfect correction can distort dynamics on timescales comparable to that of the signal drift. We note that our analysis of chemotaxis signaling dynamics presented here is relatively insensitive to such artifacts, given that the longest timescales we observed ($\sim 400$ s for sensory adaptation, *Figure 1*) are well below the time constants of fluorescence intensity drift (>1 hr under hour experimental conditions), but caution is warranted for future applications to systems with slower dynamics. More generally, quantification of single-cell variability is a challenging task because any experimental noise source can potentially contribute to the observed variability. Although we have not undertaken here a comprehensive survey of experimental noise sources for single-cell FRET, our results demonstrate meaningful differences in variability across cells measured under identical experimental conditions. The experimental duration for single-cell FRET is photon-limited, meaning that optimal experimental strategies must carefully negotiate with a finite photon budget an inherent trade-off between measurement duration, temporal resolution, and signal-to noise ratio. Future improvements of donor/acceptor fluorophores (in parameters such as photostability, brightness, maturation, as well as FRET efficiency) could enhance the effective photon budget, and hence the power of the experiment.

### From gene-expression noise to diversity in signaling phenotypes

A key feature of bacterial chemotaxis as an experimental system is that one can study *in vivo* signaling and behavior in a manner that is decoupled from gene expression and growth. Being an entirely protein-based signaling network, chemotaxis signaling responses do not require changes in gene expression, and the relatively short timescales of signaling reactions (subsecond to minutes) are well separated from those of changes in protein counts due to gene expression noise (minutes to hours). The ensemble of single-cell FRET time series measured in each of our experiments thus provide a snapshot of cell-to-cell variability due to stochastic gene expression in a variety of signaling parameters.

**Table 1.** Variability in signaling parameters reported in this study with 95 % CI obtained by bootstrap resampling.
N.D.: Not determined; N/A: Not applicable.

| Parameter | Genotype | | | Literature |
|---|---|---|---|---|
| CheR,CheB | CheRB+ | CheRB- | CheRB- | |
| Chemoreceptors | + (all) | + (all) | Tsr+ | |
| $CV(a_0)$ | $0.23 \pm 0.06$ | N/A | N/A | |
| $CV(\tau_{\mathrm{recovery}})$ | $0.20 \pm 0.06$ | N/A | N/A | 0.18–0.5[*] |
| $CV(\Pi)$ | $0.40 \pm 0.10$ | N/A | N/A | |
| $CV(K)$ | N.D. | $0.49 \pm 0.09$ | $0.16 \pm 0.07$ | |
| $\eta$ | $0.44 \pm 0.12$ | $0.09 \pm 0.04$ | $0.49 \pm 0.09$ | >0.2[†] |
| $CV(\eta)$ | $0.52 \pm 0.08$ | $1.25 \pm 0.60$ | $0.64 \pm 0.12$ | |

[*]*Berg and Tedesco, 1975*; *Min et al., 2012*
[†]*Tu and Grinstein, 2005*
DOI: https://doi.org/10.7554/eLife.27455.030

Our data revealed high variability in important signaling parameters connected to the adaptation system (*Figure 1*). In the case of the variability in recovery times (*CV*=0.20), this is likely due to variability in the CheR/receptor ratio from cell to cell. What consequences might such variability have on chemotactic behavior? A recent theoretical study has established that long (short) adaptation times are better suited for maximizing chemotactic migration rates in shallow (steep) gradients (*Frankel et al., 2014*). Thus, variability in adaptation times could partition the population into cells that will be more efficient in running up steep gradients, and others better suited to climbing shallow ones. Interestingly, it was also found that optimal performance at each gradient involves tuning not only the adaptation time, but also other parameters such as swimming speed or tumble bias, leading to the prediction that selective pressures act not only on the distribution of individual parameters, but also on the pattern of covariation among them (*Frankel et al., 2014*; *Waite et al., 2016*). Exploring such correlated variation of signaling parameters, both under changes in environmental conditions such as nutrient levels (*Khursigara et al., 2011*) and within identically grown populations, would be a fruitful avenue for future single-cell FRET studies.

In the ligand response of the network, we observed large cell-to-cell variability in the sensitivity $(1/K)$ and steepness $(H)$ of dose-response relations, for cells with a wildtype receptor population (*Figure 2*). Using a mixed-species MWC model (*Mello and Tu, 2005*), we were able to estimate the Tar/Tsr ratio in single cells, which spans a broad range from nearly zero to more than two. This strong variability in the receptor-cluster composition has the potential to dramatically impact behavior. In their natural habitats, cells likely experience a variety chemoeffector gradients simultaneously, each associated with an unknown fitness payoff for chemotactic pursuit. Generating diversity in the chemoreceptor ratio, which has been shown to determine which gradient to climb when challenged with such conflicting possibilities (*Kalinin et al., 2010*), could allow the isogenic population to hedge its bets to maximize net fitness gains. The Tar/Tsr ratio has also been shown to play an important role in setting the preferred temperature for thermotaxis (*Salman and Libchaber, 2007*; *Yoney and Salman, 2015*; *Paulick et al., 2017*). Variability in Tar/Tsr would allow diversification of the preferred temperature across cells in the population, which will promote spreading of bacteria in environments with temperature gradients. Finally, when chemotactic bacteria colonize an initially nutrient-rich environment, they are known to successively exploit resources by emitting multiple traveling waves of chemotactic bacteria, each of which consumes and chases by chemotaxis a different nutrient component outward from the colony origin (*Adler, 1966*). Our observation that the population diversity in receptor ratios, and hence chemotactic preference, varies concomitantly with population growth could provide a means to tune the population fractions that engage in such excursions into virgin territory, and those that remain for subsequent exploitation of remaining resources. Thus, the diversity in ligand response and preference generated by variability in the Tar/Tsr ratio could have

nontrivial consequences in a variety of behavioral contexts encountered by isogenic chemotactic (and thermotactic) populations.

## Suppression of gene expression noise by CheB phosphorylation feedback

The role of phosphorylation feedback has been a long standing open question in the field of bacterial chemotaxis signaling, ever since its presumed role in providing precise adaptation was decisively ruled out by (*Alon et al., 1999*). In the ensuing years, a diverse set of hypotheses have been proposed to explain its purpose. Apart from precise adaptation, CheB phosphorylation has been suggested as possibly responsible for the non-linear response of CheB activity to changes in CheA kinase activity (*Shimizu et al., 2010*; *Clausznitzer et al., 2010*), ligand sensitivity of wildtype cells (*Barkai et al., 2001*), and has been implicated theoretically as a possible mechanism to buffer gene-expression noise to suppress detrimental variability in the steady-state kinase activity (*Kollmann et al., 2005*; *Emonet and Cluzel, 2008*; *Pontius et al., 2013*). Here, we tested the latter hypothesis, by severing the phosphorylation feedback loop as a possible noise-reduction mechanism. Our single-cell FRET data revealed that, not only does CheB phosphorylation feedback strongly attenuate the magnitude of variability in the steady-state kinase activity $a_0$, it also qualitatively changes the shape of the distribution $P(a_0)$ across cells to convert an otherwise bimodal distribution into a unimodal one (*Figure 3d*). The highly polarized bimodal distribution of steady-state activities in CheB phosphorylation mutants are likely detrimental, as they could drive $a_0$ of a large fraction of the population too far from the flagellar motor's steep response threshold (*Cluzel et al., 2000*; *Yuan and Berg, 2013*) to effectively control swimming.

By analyzing simplified models of adaptation kinetics, we found that a bimodal $P(a_0)$ could occur in the absence of phosphorylation feedback if the enzyme kinetics of CheR and CheB depend sublinearly on the activity $a$ of their receptor substrates. As a limiting case, when both enzymes work at or near saturation, this model leads to zero-order ultrasensitivity (*Goldbeter and Koshland, 1981*; *Emonet and Cluzel, 2008*), which could act as a strongly non-linear transfer function $f([R]/[B])$ that converts a unimodal distribution $P([R]/[B])$ into a bimodal $P(a_0)$. We note that ultrasensitivity due to sublinear (Michaelis-Menten) enzyme kinetics is by no means the only possible explanation for the observed bimodality in $P(a_0)$. Any mechanism that renders $f([R]/[B])$ a strongly nonlinear (sigmoidal) function could lead to the same effect. The merit of the sublinear kinetic (ultrasensitivity) model is in its simplicity, but it is worth noting that reality is likely to be more complex due to, for example, effects of spatial organization. It is known that both CheR and CheB interact with chemoreceptors not only at their substrate modification residues, but also with a second binding site on a flexible tether at the receptor C-terminus. Such bivalent interactions with the receptor array could affect the movement of these enzymes across the receptor lattice (*Levin et al., 2002*), and such movements could shift the balance between processivity and distributivity of enzyme activity on their substrate receptors (*Pontius et al., 2013*), which could in turn attenuate or enhance the nonlinearity in the relationship $f([R]/[B])$ between the enzyme expression ratio [R]/[B] and the steady-state activity $a_0$ of their substrate receptors (*Takahashi et al., 2010*).

## Diversity in temporal variability: bet-hedging across exploration and exploitation strategies

In addition to cell-to-cell variability in signaling parameters, single-cell FRET allowed us to resolve temporal fluctuations in signaling about the steady-state output within individual cells. In wildtype cells, we found that the steady-state activity fluctuates slowly (*Figure 4*, correlation time $\tau \approx$ 10s) with a large amplitude ($\eta = \sigma_a/\langle a \rangle \approx$ 40%), but this amplitude also varies significantly from cell to cell ($CV \approx 0.6$). Fluctuations on this timescale were absent in CheRB- cells defective in receptor methylation/demethylation, indicating that these fluctuations are generated by stochastic processes in the activity of the adaptation enzymes CheR and CheB. Whereas the fluctuation correlation time $\tau$ in our FRET experiments was in close agreement with those from previously reported flagellar motor switching experiments (*Korobkova et al., 2004*; *Park et al., 2010*), the fluctuation amplitude $\langle \eta \rangle \approx$ 40% was surprisingly large. Theoretical analysis of the motor-based noise measurements indicated that, in the frequency range of our experiments, stochastic methylation kinetics are indeed the dominant source of noise (*Clausznitzer and Endres, 2011*). Another theoretical study of the motor

noise (*Tu and Grinstein, 2005*), had predicted a modest noise level of intracellular noise, with a lower bound of 20% of the mean. The discrepancy is likely due, at least in part, to the recently discovered adaptation at the level of the flagellar motor (*Yuan et al., 2012*), which must effectively act as a high-pass filter that attenuates frequencies near or below a cutoff frequency determined by its own characteristic timescale for adaptation. The fluctuation amplitude $\eta$ was also much greater than previous estimates from pathway-based models and we have shown that there is an additional noise source, independent from methylation, which contributes to the total noise amplitude in wildtype cells and not considered in previous modeling efforts.

The large temporal noise we observed in wildtype (CheRB+) cells may seem counterintuitive, given that the chemotaxis pathway is a transduction path for sensory information, and noise generally reduces information transmission capacity of communication channels (*Shannon, 1949*). However, the chemotaxis signaling pathway is not only a sensory system but also a control circuit for motile behavior, and recent studies have highlighted the importance of considering the behavioral context in understanding the design of this signaling pathway (*Dufour et al., 2016*; *Wong-Ng et al., 2016*; *Long et al., 2017*). The temporal noise we observed could have profound implications for *E. coli*'s random-walk motility strategy, because slow fluctuations in the intracellular signal can enhance the likelihood of long run events and stretch the tail of the run-length distribution to yield power-law-like switching-time distributions over a range of time scales (*Korobkova et al., 2004*; *Tu and Grinstein, 2005*). Such non-exponential statistics are known to yield superior foraging performance in environments where resource distribution is sparse (*Viswanathan et al., 1999*), and temporal fluctuations in run-tumble behavior has also been shown theoretically to enhance climbing of shallow gradients by generating runs that are long enough to integrate over the faint gradient a detectable difference in ligand input (*Flores et al., 2012*; *Sneddon et al., 2012*). Hence, the noise generated by the adaptation system can be advantageous in resource-poor environments (*deserts*) in which efficient exploration of space for sparsely distributed sources (*oases*) is of utmost importance. By contrast, strong temporal noise clearly degrades response fidelity in rich environments where the gradient signal is strong enough for detection with short runs, and might also complicate coordination of cells in collective behaviors such as the aforementioned traveling-wave exploitation of nutrients. Our finding that the noise amplitude varies strongly from cell to cell thus suggests that isogenic populations might be hedging their bets by partitioning themselves between specialists for local exploitation of identified resource patches and those for long-range exploration in search for new ones.

## Giant fluctuations and digital switching in adaptation deficient cells with homogeneous chemoreceptor arrays

We found the most dramatic temporal fluctuations in adaptation-deficient (CheRB-) cells expressing Tsr as the sole chemoreceptor species (*Figure 5*). When brought close to their dose-response transition point ($K$) by attractant stimulation, these cells demonstrated strong temporal fluctuations, revealing that there exist sources of signal fluctuations that are independent of CheR and CheB activity. The origin of these adaptation-independent fluctuations remain unknown, but in broad terms, one can envisage that they are due to either intrinsic sources (i.e. fluctuations arising within the components of the receptor-kinase complex), extrinsic sources (i.e. fluctuations in other cellular processes/environmental variables), or both. Possible intrinsic sources include coupled fluctuations in protein conformations (*Duke and Bray, 1999*; *Shimizu et al., 2003*; *Mello et al., 2004*; *Skoge et al., 2011*), the slow-timescale changes in receptor 'packing' that have been observed by fluorescence anisotropy measurements (*Frank and Vaknin, 2013*; *Vaknin, 2014*), and the stochastic assembly dynamics of receptor clusters (*Greenfield et al., 2009*). Possible extrinsic sources include fluctuations in metabolism, membrane potential, or active transport/consumption of ligand. Many of these possibilities could be tested by experiments of the type presented here with appropriate mutant strains and environmental controls, and present promising directions for future research.

The adaptation-independent fluctuations we observed were not only large in amplitude but often (though not always) took the form of discrete steps in activity, in some cases between only two levels. Two-state descriptions of receptor signaling are a common feature of nearly all mechanistic models of bacterial chemotaxis signaling addressing both cooperativity (*Duke and Bray, 1999*; *Shimizu et al., 2003*; *Mello et al., 2004*; *Mello and Tu, 2005*; *Keymer et al., 2006*) and adaptation (*Asakura and Honda, 1984*; *Barkai and Leibler, 1997*; *Morton-Firth et al., 1999*; *Endres and*

*Wingreen, 2006*; *Emonet and Cluzel, 2008*; *Tu et al., 2008*), yet direct evidence for two-state switching by receptor-kinase complexes has been lacking. Although as noted above, it is yet possible that the two-level switching we observed (*Figure 5d*) is due to extrinsic noise sources (e.g. metabolism or transport), the temporal statistics (*Figure 5e–h*) are compatible with a simple model in which two stable signaling states are separated by an energy barrier sensitive to both environmental stimuli and internal cell variables. Regarding cells that exhibited step-like transitions among more than two stable states, a plausible interpretation is that the underlying transitions are actually two-level, but the majority of the receptor-kinase population is partitioned into two or more disjoint signaling arrays which fluctuate independently.

While two-state switching has been observed in small oligomers such as ion channels (*Keller et al., 1986*) and larger protein assemblies such as the bacterial flagellar motor (*Silverman and Simon, 1974*; *Bai et al., 2010*), controlled by up to a few dozen units, our findings suggest (as discussed in results) that at least many hundreds, if not thousands of receptor-kinase units can switch in a concerted fashion. The rather long timescale associated with intervals between switches ($\approx 10^2$ s) is clearly distinct from the methylation-dependent fluctuation timescale ($\approx 10^1$ s) observed in CheRB+ cells, and might reflect the large size of the cooperatively switching signaling array. The switching duration ($\approx 4$ s), is also much slower than the sub-second response to attractant stimuli (*Segall et al., 1982*; *Sourjik and Berg, 2002b*). These fluctuations of surprisingly large magnitude indicate the possibility that cooperativity between arrayed chemoreceptors are much stronger than suggested by previous population-averaged measurements, and represent a promising direction for future investigations.

## Concluding remarks

We described a new single-cell FRET technique capable of resolving intracellular signaling dynamics in live bacteria over extended times. Our results highlight how a protein-based signaling network can either generate or attenuate variability, by amplifying or filtering molecular noise of different molecular origins. Gene expression noise is harnessed, on the one hand, to generate diversity in the ligand response of isogenic populations, or attenuated, on the other the hand, in the control of steady-state signal output. In addition, we showed that signaling noise generated at the level of interacting gene products can have a profound impact. Stochastic protein-protein interactions within the signaling network, as well as other 'extrinsic' fluctuations, can be amplified by the signaling network to generate strong temporal fluctuations in the network activity.

# Materials and methods

## Strains and plasmids

All strains used are descendants of *E. coli* K-12 HCB33 (RP437). Growth conditions were kept uniform by transforming all strains with two plasmids. All strains and plasmids are shown in *Tables 2* and *3*.

The FRET acceptor-donor pair (CheY-mRFP and CheZ-YFP) is expressed in tandem from a IPTG inducible pTrc99A plasmid, pSJAB12 or pSJAB106, with respective induction levels of 100 and 50 µM IPTG. The differences between pSJAB12 and pSJAB106 are (i) the presence of a noncoding spacer in pSJAB106 to modify the ribosome binding site of CheZ (*Salis et al., 2009*), such that CheZ is expressed approximately three fold less, and (ii) a A206K mutation in YFP to enforce monomerity. We also used pVS52 (CheZ-YFP) and pVS149 (CheY-mRFP1) to express the fusions from separate plasmids with induction levels of 50 µM IPTG and 0.01 % arabinose, respectively. We transformed the FRET plasmids in an adaptation-proficient strain (VS104) to yield CheRB+ and an adaptation-deficient strain (VS149) to get CheRB-. For attachment with sticky flagella from pZR1 we used the equivalent strains in *fliC* background (VS115 and TSS58).

Experiments with Tsr as the sole chemoreceptor were performed in UU2567 or TSS1964, in which the native FliC gene is changed to sticky FliC (FliC*). Tsr is expressed from pPA114 Tsr, a pKG116 derivative, at with an induction of 0.6 µM NaSal.

For the experiments with the CheB mutants, pSJAB12 was transformed into VS124 together with plasmids expressing CheB$^{WT}$, CheB$^{D56E}$ and truncated mutant CheB$_c$ (plasmids pVS91, pVS97 and pVS112, respectively, with induction levels of 1.5E-4, 6E-4 and 3E-4 % arabinose.

## FRET microscopy

Föster Resonance Energy Transfer [FRET] microscopy was performed as previously reported (*Sourjik et al., 2007*; *Vaknin and Berg, 2004*). Cells were grown to OD = 0.45–0.5 in Tryptone Broth (TB) medium from a saturated overnight culture in TB, both with 100 g/mL ampicillin and 34 g/mL chloramphenicol and appropriate inducers in the day culture. For the FRET experiments we used Motility Media (MotM, (*Shimizu et al., 2006*)), in which cells do not grow and protein expression is absent. Cells were washed in 50 mL MotM, and then stored 0.5–6 hr before experiment. In the dose-response curve experiments and the temporal fluctuation measurements, cells were stored up to three hours at room temperature to allow for further red fluorescence maturation. A biological replicate or independent FRET experiment was defined as a measurement from separately grown cultures, each grown on a separate day.

Cells were attached by expressing sticky FliC (FliC*) from a pKG116 plasmid or the chromosome (TSS1964), induced with 2μM Sodium Salicylate (NaSal), or with Poly-L-Lysine (Sigma), or with anti-FliC antibodies column purified (Using Protein A sepharose beads, Amersham Biosciences) from rabbit blood serum and pre-absorbed to FliC- cells (HCB137, gifts from Howard Berg). We found FRET experiments with sticky FliC to have the highest signal-to-noise ratio.

Fluorescent images of the cells were obtained with a magnification of 40-100x (Nikon instruments). For excitation of YFP, we either used 514 nm laser excitation set to 30 mW for 2 ms or an LED system (CoolLED, UK) with an approximate exposure time of 40 ms to approximate the same illumination intensity per frame. The sample was illuminated stroboscopically with a frequency between 1 and 0.2 Hz. RPF excitation was performed by 2 ms exposure of 60 mW 568 nm laser or equivalent with LED to measure acceptor levels independently from FRET.

Excitation light was sent through a 519 nm dichroic mirror (Semrock, USA). Epifluorescent emission was led into an Optosplit (Cairn Research, UK) with a second dichroic mirror 580 nm and two emission filters (527/42 nm and 641/75 nm, Semrock, USA) to project the RFP and YFP emission side by side on an EM-CCD (Princeton Instruments, USA) with multiplication gain 100.

## Image processing

Images were loaded and analyzed by means of in-house written scripts (Image segmentation script FRETimaging.py available online) in MATLAB and Python. For ratiometric FRET experiments, we segmented single cells using the donor emission with appropriate filter steps to remove clusters of cells or cells improperly attached to the coverslip. At the position of each cell a rectangular ROI is defined in which all fluorescence intensity is integrated.

**Table 2.** Strains used in this study.

| Background | | | Plasmids | |
| --- | --- | --- | --- | --- |
| Strain | Source | Relevant genotype | Plasmid 1 | Plasmid 2 |
| VS115 | V. Sourjik | ΔYZ ΔFliC | pSJAB106 | pZR1 |
| VS104 | *Sourjik and Berg, 2002a* | ΔCheYZ | pSJAB12 | pBAD33 |
| TSS58 | this work | ΔRBYZ ΔFliC | pSJAB106 | pZR1 |
| VS149 | *Sourjik and Berg, 2004* | ΔRBYZ | pVS12 | pVS33 |
| VS124 | *Clausznitzer et al., 2010* | ΔCheBYZ | pSJAB12 | pVS112 |
| VS124 | | ΔCheBYZ | pSJAB12 | pVS97 |
| VS124 | | ΔCheBYZ | pSJAB12 | pVS91 |
| UU2567 | *Kitanovic et al., 2015* | ΔCheRBYZ,ΔMCP[†] | pSJAB106 | pPA114 Tsr |
| TSS1964 | this work | ΔCheRBYZ,ΔMCP FliC* | pSJAB106 | pPA114 Tsr |
| UU2614 | J.S. Parkinson | CheB Δ(4-345) | pTrc99a | pVS91,97,112 |

All strains are descendants of E. coli K-12 HCB33 (RP437). In all FRET experiments, strains carry two plasmids and therefore confer resistance to chloramphenicol and ampicillin.

[†]all five chemoreceptor genes *tar tsr tap trg aer* deleted.

*expresses sticky FliC filament (*Scharf et al., 1998*)

DOI: https://doi.org/10.7554/eLife.27455.031

**Table 3.** Plasmids used in this study.

| Plasmid | Product | System | Ind | Res | Source |
|---|---|---|---|---|---|
| pVS52 | CheZ-5G-YFP | pBAD33 | ara | cam | *Sourjik and Berg, 2002a* |
| pVS149 | CheY-5G-mRFP1 | pTrc99a | IPTG | amp | *Sourjik and Berg, 2002a* |
| pSJAB12 | CheZ-5G-YFP/CheY-5G-mRFP1 | PTrc99a | IPTG | amp | This work |
| pSJAB106 | CheZ-5G-YFP/CheY-5G-mRFP1[¶] | PTrc99a | IPTG | amp | This work |
| pVS91 | CheB[†] | pTrc99a | ara | cam | *Liberman et al., 2004* |
| pVS97 | CheB-D56E[‡] | pBAD33 | ara | cam | *Clausznitzer et al., 2010* |
| pVS112 | CheBc[§] | pBAD33 | ara | cam | V. Sourjik |
| pSJAB 122 | CheBc-GS4G-mVenus | pBAD33 | ara | cam | This work |
| pSJAB 123 | CheB(D56E)-GS4G-mVenus | pBAD33 | ara | cam | This work |
| pSJAB 124 | CheB-GS4G-mVenus | pBAD33 | ara | cam | This work |
| pZR 1 | FliC* | pKG116 | NaSal | cam | This work |
| pPA114 Tsr | Tsr | pPA114 | NaSal | cam | *Ames et al., 2002* |

[¶]Contains a A206K mutation to enforce monomerity..

[†]expresses WT CheB.

[‡]carries a point mutation D56E in CheB.

[§]expresses only residues 147–349 of CheB, preceded by a start codon (Met).

*expresses sticky FliC filament (*Scharf et al., 1998*)

DOI: https://doi.org/10.7554/eLife.27455.032

For FRET experiments in which the concentration of donor molecules may influence the FRET signal, the experiments on the CheB mutants, segmentation was done separately for each frame to determine the cell shape and then linking these segmented images with a tracking algorithm (*Crocker and Grier, 1996*), afterwards, fluorescence intensities are normalized for the cell size (mask surface area) in segmentation, intensities were corrected for inhomogeneous illumination, and cells with low acceptor intensities were excluded from the analysis. The ROI for the donor intensity were subsequently used to obtain the acceptor intensity per cell, both in photon-count per pixel.

Fluorescence intensities were corrected for long-time drift (primarily due to bleaching, but also including contributions from fluorophore maturation and/or recovery from long-lived dark states) by fitting a linear, single exponential or double exponential function to the fluorescence decay, separately for both donor and acceptor channels. The net decay in the FRET signal was dominated by photobleaching of the donor (YFP) intensity (on average 25% over the course of a 30 min experiment; *Figure 1—figure supplement 2*). Red fluorescent proteins tend to have long maturation times, which under our experimental conditions (in which gene expression is halted upon harvesting via auxotrophic limitation) could result in a residual increase in red fluorescence intensity during experiments. In control experiments, we determined mRFP1's maturation half time under our conditions to be ~2–3 hr, with a maximum increase in the FRET signal of ~25% at ~5–6 hr. Cells in which the intensity decay could not accurately be corrected were excluded from the analysis.

**Table 4.** List of global parameters used for model of Mello and Tu.

In these fits, $\tilde{K}$ is a free parameter while others are constrained ±5% by published values.

| Parameter | Start value (*Mello and Tu, 2007*) | Final value |
|---|---|---|
| $C$ | 0.314 | 0.29 |
| $\epsilon_0$ | 0.80 | 0.84 |
| $\epsilon_A$ | 1.23 | 1.29 |
| $\epsilon_S$ | 1.54 | 1.61 |
| $\tilde{K}$ | – | 21.2 µM |

DOI: https://doi.org/10.7554/eLife.27455.033

In non-ratiometric fluorescence experiments (CheB-mVenus) the fluorescence intensities obtained after segmentation were corrected for inhomogeneous illumination and divided by cell area.

## FRET analysis

The FRET signal is calculated from fluorescent time series. We observe changes in the ratio $R = A/D$, in which A and D are the fluorescence intensities of the acceptor and donor. In previous population-averaged FRET experiments the FRET per donor molecule ($\Delta D/D_0$) is calculated as (*Sourjik and Berg, 2002a*; *Sourjik et al., 2007*):

$$\frac{\Delta D}{D_0} = \frac{\Delta R}{\alpha + R_0 + \Delta R} \tag{4}$$

in which $R_0$ is the ratio in absence of FRET, $\alpha = |\Delta A/\Delta D|$ is a constant that depends on the experimental system (in our case $\alpha = 0.30$) and the change in ratio as a result of energy transfer $\Delta R$ and $R_0$ are obtained through observing the ratio just after adding and removing saturated attractant stimuli. This expression is convenient for population FRET since is invariant to attachment densities of a population. However, in single-cell FRET this expression may generate additional variability in FRET due to variable donor levels from cell to cell. Hence it is more convenient to define the FRET levels in terms of the absolute change in donor level $\Delta D$, since this reflects the number of resonance energy transfer pairs

$$\text{FRET}(t) = \Delta D = D_0 \frac{\Delta R}{\alpha + R_0 + \Delta R} \tag{5}$$

Since FRET occurs only when CheY-P and CheZ interact, the FRET level is proportional to the concentration of complex [Yp-Z]. If we assume the CheY-P dephosphorylation by CheZ follows Michaelis-Menten kinetics we can describe the [Yp-Z] concentration in terms of the activity of the kinase CheA. For this, we assume the system is in steady-state for timescales much larger than CheY phosphorylation-dephosphorylation cycle ($\approx 100$ ms). In that case, the destruction rate should equal the rate of CheA phosphorylation and hence the FRET signal is proportional to the activity per kinase $a$ and the amount of CheA in the receptor-kinase complex (*Sourjik and Berg, 2002a*; *Oleksiuk et al., 2011*):

$$\text{FRET} \propto [\text{Yp} - \text{Z}] = a \frac{k_A}{k_Z}[\text{CheA}] \approx a \frac{k_A}{k_Z}[\text{CheA}]_\text{T} \tag{6}$$

This last step is only valid if we further assume CheA autophosphorylation being the rate-limiting step. This is the case only if sufficient amounts of CheZ and CheY present in the cell. We have found that the FRET level initially increases with donor (CheZ) levels, but then saturates and remains constant for CheY and CheZ (see *Figure 3—figure supplement 2*).

In many cases the most relevant parameter is the normalized FRET response. The FRET level reaches maximum if all kinases are active ($a \approx 1$). In case of CheRB+cells, this is the case when removing a saturating amount of attractant after adaptation (*Sourjik and Berg, 2002a*). For CheRB- cells the baseline activity is (*Sourjik and Berg, 2002a*; *Shimizu et al., 2010*) close to 1. Hence the normalized FRET $\text{FRET}(t)/\text{FRET}_\text{max}$ represents the activity per kinase $a(t)$ and is the relevant parameter for many quantitative models for chemoreceptor activity (*Tu, 2013*).

$$a(t) = \frac{\text{FRET(t)}}{\text{FRET}_{max}} \tag{7}$$

and from $a(t)$ the steady-state activity $a_0$ can be determined by averaging $a(t)$ over baseline values before adding attractant stimuli.

## Analysis of power spectra

From FRET time series of length $T$ and acquisition frequency $f$ we calculated Power Spectral Density (PSD) estimates as

$$S_{\text{FRET}}(\omega) = \frac{1}{T}|\mathscr{F}(\omega)|^2 \tag{8}$$

where $\mathscr{F}(\omega)$ is the (discrete-time) Fourier transform of the FRET time series $\mathrm{FRET}(t)$. We only consider positive frequencies and multiply by two to conserve power.

To study the influence of experimental noise and the effect of estimating $\tau$ and $c$ from a finite time window, we generated O-U time series using the update formula (*Gillespie, 1996*)

$$X(t+\Delta t) = X(t) - \tau^{-1}\Delta t + c^{1/2}n(\Delta(t))^{1/2} \tag{9}$$

in which $n$ denotes a sample value from a normally distributed random variable ($\mu = 0, \sigma = 1$). To the generated time series Gaussian white noise was added to simulate experimental noise. The experimental noise amplitude was obtained from the average power at high frequencies.

## Two-state switching analysis

Since the amplitude of two-state switches is much greater than the noise, switching events times $t_0$ could be easily read off by eye. We obtained switching durations by fitting the function

$$a(t) = \frac{1}{2} \pm \frac{1}{2}\frac{t-t_0}{|t-t_0|}(1-e^{-2|t-t_0|/\tau_\pm}) \tag{10}$$

to the normalized FRET time series in a 30 s time window, approximately $\pm 15$ s from $t_0$. The residence times $\Delta t_{\mathrm{up},i,k}$ and $\Delta t_{\mathrm{down},i,k}$ of event $k$ in cell $i$ were defined by the time between transitions or the beginning/end of the 25 μM stimulus time window. The steady-state activity during activity was then calculated as

$$a_{1/2,i} = \frac{\sum_k \Delta t_{\mathrm{up},i,k}}{\sum_k \Delta t_{\mathrm{down},i,k} + \sum_k \Delta t_{\mathrm{up},i,k}} \tag{11}$$

and for the residence times we take the mean over $k$ to get $\tau_{\mathrm{down}}$ and $\tau_{\mathrm{up}}$. If we treat the system as an equilibrium process we can use the Arrhenius equations that describe the residence times as a function of the distance to the energy barrier

$$\begin{aligned}
\tau_{\mathrm{down}} &= \tau_r \exp\left[\gamma_{\mathrm{down}}\ln\left[a_{1/2}/(1-a_{1/2})\right]/k_B T\right] \\
\tau_{\mathrm{up}} &= \tau_r \exp\left[\gamma_{\mathrm{up}}\ln\left[a_{1/2}/(1-a_{1/2})\right]/k_B T\right]
\end{aligned} \tag{12}$$

in which $\gamma_{\mathrm{down}}$ and $\gamma_{\mathrm{up}}$ are constants corresponding to the slopes of $\ln\tau_{\mathrm{down}}$ and $\ln\tau_{\mathrm{up}}$ against $\ln\left[a_{1/2}/(1-a_{1/2})\right]$, respectively. The fit parameters and standard error are obtained with the robustfit function in Matlab (statistics toolbox).

## Dose response curve analysis

Normalized FRET responses to different levels of ligand are fit to a hill curve of the form

$$a = \frac{[L]^H}{[L]^H + [K]^H} \tag{13}$$

This can be connected to an MWC-type model (*Monod et al., 1965*) of receptor cluster activity (*Tu et al., 2008*) in the regime $K_I \ll [L] \ll K_A$, resulting in the correspondence key

$$\begin{aligned}
H &= N \\
K &= K_I e^{f_m(m)}
\end{aligned}$$

which relates the Hill slope directly to the cluster size $N$, and sensitivity $K$ to the methylation energy of the receptor. We plot $K$ on a logarithmic scale to scale linearly with energy.

The parameter estimate uncertainties of $K$ and $H$ are defined by the covariance matrix for each cell $i$

$$\mathrm{COV}_i = \begin{bmatrix} \sigma_{KK} & \sigma_{HK} \\ \sigma_{KH} & \sigma_{HH} \end{bmatrix}_i \tag{14}$$

in which $\sigma$ denotes the standard error from the fit. For each covariance matrix the corresponding

eigenvectors and eigenvalues are determined. The eigenvalues and vectors constitute an ellipsoid which represent error basins in $K - H$ space.

To obtain expression level estimates of different receptor species we use a different MWC model. Following (*Mello and Tu, 2005*), we use as an expression for the normalized response of cells to ligand $[L]$ serine

$$a = \frac{\epsilon_0 \epsilon_S^{N_S} \epsilon_A^{N_A} (1 + C[L]/\tilde{K})^{N_s}}{(1 + [L]/\tilde{K})^{N_s} + \epsilon_0 \epsilon_S^{N_S} \epsilon_A^{N_A} (1 + C[L]/\tilde{K})^{N_s}} \tag{15}$$

in which $N_A$ is the number of Tar receptors in the cluster and $N_S$ is the number of Tsr receptors. Parameters $\epsilon_A$, $\epsilon_S$, $\epsilon_0$ are the energies corresponding to binding of ligand to Tar, Tsr and the other three receptors and are the same for each cell, like $C$ and $\tilde{K}$ which describe the disassociation constant for the active state as $K_A = \tilde{K}/C$, while $N_A$ and $N_T$ may vary from cell to cell. This yields the minimization problem for all 128 cells

$$\min \sum_i^{N_{cells}} \sum_j^{N_{stim}} (m_{i,j} - a_{i,j})^2 \tag{16}$$

in which $m_{i,j}$ the measured FRET response normalized to the response amplitude of cell $i$ to stimulus $L_j$. This function was minimized using the matlab function fmincon (optimization toolbox). The total number $N_T = N_A + N_S$ is limited to 32. When fitting the model used the energy parameters $\epsilon$ from reference (*Mello and Tu, 2005*) where used as initial guess with a maximum of $\pm 5\%$ deviation. This yielded an estimate of $N_A$ and $N_S$ for each cell. Under the assumption that receptor clusters are well-mixed, this yields a Tar/Tsr ratio of $N_A/N_S$.

## Sublinear model of adaptation kinetics with phoshorylation feedback

For our model, we consider CheR and CheB to perform opposite operations on the same substrate. For simplicity, we do not explicitly describe the methylation and demethylation of the receptors explicitly but instead assume that CheR (R) activates the receptor-kinase complex directly (A*), and that CheB (B) deactivates it (A)

In general, the corresponding reaction equation is a function of the methylation of inactive kinases by CheR, and demethylation of active kinases by CheB, described by two functions $g$ and $h$

$$\frac{da}{dt} = g(v_r, a) - h(v_b, a) \tag{17}$$

with $v_r$ and $v_b$ being the rates for CheR and CheB, respectively. We now assume that these reactions follow Michaelis-Menten kinetics, following (*Goldbeter and Koshland, 1981*) and (*Emonet and Cluzel, 2008*), and the total amount of kinase complexes is constant ($A_T = A^* + A$). Hence the change in activity $a = A^*/A_T$ has a sublinear dependence on $a$:

$$\frac{da}{dt} = v_r \frac{1 - a}{K_r + 1 - a} - v_b \frac{a}{a + K_b} \tag{18}$$

The Michaelis-Menten constants $K_b$ and $K_r$ are in units of $A_T$ and are therefore dimensionless numbers. We are interested in the steady-state level $a_0$ and its dependence on the kinetic parameters in *equation 18*. This is described by the Goldbeter-Koshland function (*Tyson et al., 2003*), an exact solution to the system in case [R] and [B] are much smaller than $[A]_T$.

$$a_0[v_r, v_b, K_r, K_b] = \frac{2 v_r K_r}{(v_b - v_r + v_b K_b + v_r K_r) + \sqrt{(v_b - v_r + v_b K_b + v_r K_r)^2 - 4(v_b - v_r) v_r K_r}} \tag{19}$$

The shape of this curve is sigmoidal if the Michaelis-Menten constants $K_r$ and $K_b$ are much smaller than one. For CheB phosphorylation, we assume the phosphorylation rate depends linearly on active CheA and write

$$\frac{d[\text{Bp}]}{dt} = k_p [\text{B}] a(v_r, v_b, K_b, K_r) - k_{dp}[\text{Bp}] \tag{20}$$

with the corresponding conservation law $B_T = B_P + B$. For the case for wild-type CheB, with phosphorylation feedback, the rates can be described in terms of catalytic rate times the enzyme (subspecies) concentration

$$\begin{aligned} v_b &= k_b([\mathrm{B}_T] - [\mathrm{Bp}]) + M k_b [\mathrm{Bp}] \\ v_r &= k_r[\mathrm{R}] \end{aligned} \tag{21}$$

in which $M$ stands for the ratio of demethylation rates of unphosphorylated and phosphorylated CheB. The fraction of the phosphorylated CheB, $[\mathrm{Bp}]/[\mathrm{B}]_\mathrm{T}$ then determines the effective activity of CheB. *Equation 20* is solved numerically using Mathematica (Mathematica model source code available online) for [Bp] and the result is substituted in *equation 19*. In the absence of feedback, the activity can be directly calculated from *equation 19* with the rates being simply

$$\begin{aligned} v_b &= k_b[\mathrm{B}] \\ v_r &= k_r[\mathrm{R}] \end{aligned}$$

We only need to consider the ratio of rate constants $k_r$ and $k_b$ which determines at which expression ratio [CheR]/[CheB] the activity equals 1/2. We assume $k_r = k_b$ for simplicity, since the shape of the curve from *Equation 19* is not affected by the values of $k_r$ and $k_b$, changing their ratio only shifts the curve along the horizontal axis. Similarly, we only consider the ratio of phosphorylation and dephosphorylation rates. This leaves the system of equations above only has a few parameters: $K_{b,r}$ and $M$; and the ratios $k_r/b_b$ and $k_p/k_{dp}$. In *Table 5*, the parameters used for the calculations are listed.

We first fixed the phosphorylation rates $k_p = 1/2 k_{dp}$. This means that the steady-state phosphorylated level of CheB $[\mathrm{B}_p]/[\mathrm{B}_T]$ at activity $\approx 1/3$ is around 15 %. This parameter is not constrained by any direct observation, but it is clear the system benefits from a relatively low fraction of phosphorylation, to be able to up and down regulate the levels effectively upon changes in activity.

Generally, we assume CheB-D56E to behave like unphosphorylated CheB. The gain in catalytic rate of activated CheB is estimated to be nearly a 100 fold, but this does not agree with the expression level differences between the different CheB mutants so we made a conservative estimate of 15 (the attenuating effect increases with the gain). CheBc behaves approximately like phosphorylated CheB (albeit with increase of only seven compared to D56E), qualitatively consistent with measured in vitro rates for CheBc and phosphorylated intact CheB (*Anand and Stock, 2002*). The difference between predicted rates and might be due to the fact that the rate experiments were performed in vitro. Michaelis Menten constants used in the model are lower than 1, but how low is not well constrained by data, and estimations do not take into account the possible attenuating effect of phosphorylation. Our experimental data on the distribution of $a_0$ implies the sigmodial curve is steep in the absence of phosphorylation and hence that $K_b$ and $K_r$ are quite small. The variability in $a_0$ for CheBc is lower than D56E, implying that the curve is less steep and hence we have chosen are $K_r$ which is not quite as low as D56E.

To simulate gene expression noise, we simulated [CheR]/[CheB] log-normal distributions with $\sigma = 0.18$ for all three strains. The mean of the distribution was chosen to yield an average steady-state

**Table 5.** List of parameters used for Goldbeter-Koshland description of CheB phosphorylation feedback.

| Parameter | Value | Literature | Source |
|---|---|---|---|
| $k_r/k_p$ | 1 | 0.75 | *Shimizu et al., 2010* |
| $k_{dp}/k_p$ | 2 | $k_p = 0.37\mathrm{s}^{-1}$ | *Kentner and Sourjik, 2009* |
| $K_r$ | 0.03 | <<1 | *Emonet and Cluzel, 2008* |
| $K_b$ | 0.03 | <<1 | *Emonet and Cluzel, 2008* |
| $K_b(CheBc)$ | 0.2 | <<1 | |
| M (WT) | 15 | 100 | *Anand and Stock, 2002* |
| M (CheBc) | 7 | 15 | *Simms et al., 1985* |

DOI: https://doi.org/10.7554/eLife.27455.034

network activity ($a_0$) of 0.4. The resulting distribution of $a_0$ was calculated using the corresponding Goldbeter-Koshland function for each genotype.

## Linear and supralinear models of adaptation kinetics

Instead of assuming a sub-linear (Michaelis-Menten) dependence of CheR- and CheB-catalyzed rates on the receptor-kinase activity $a$, one may also assume linear, quadratic or cubic dependence of the methylation rates on the activity, as was for example done in (*Clausznitzer et al., 2010*). Here, CheR feedback is assumed to be linear ($g = k_r[\mathrm{R}](1-a)$), while CheB feedback can be linear ($h = k_b[\mathrm{B}]a$), quadratic ($h = k_b[\mathrm{B}]a^2$) or cubic ($h = k_b[\mathrm{B}]a^3$) in the receptor-kinase activity $a$. The supralinear (quadratic and cubic) forms of dependence are intended to model the case with CheB phosphorylation, and the linear form the case without CheB phosphorylation. The steady-state activity $a_0$ can be found by solving $g(v_r, a) = h(v_b, a)$ and the dependence of $a_0$ to [R]/[B] ($a_0$=f([R]/[B])) for these linear and supralinear cases are shown in *Figure 3—figure supplement 3*.

## Acknowledgements

We thank Sandy Parkinson and Germán Piñas for strains, many useful discussions and critical reading of the manuscript, Istvan Kleijn, Yuki Esser, Iwan Vaandrager, Francesca van Tartwijk and Pieter de Haan for help with experiments at various phases of the project, William Pontius for useful discussions, Simone Boskamp and Zuzana Rychnavska for cell culture and cloning, Marco Kamp for microscopy assistance and Marco Konijnenburg, Brahim Ait Said, Luc Blom and Eric Clay for software and electronic support. This work was supported by NWO/FOM and the Paul G. Allen Family Foundation. Thierry Emonet acknowledges support from NIH grant R01GM106189.

## Additional information

### Funding

| Funder | Grant reference number | Author |
|---|---|---|
| Paul G. Allen Family Foundation | 11562 | Thierry Emonet<br>Thomas S Shimizu |
| National Institutes of Health | R01GM106189 | Thierry Emonet |
| Nederlandse Organisatie voor Wetenschappelijk Onderzoek | NWO Vidi 680-47-515 | Thomas S Shimizu |
| Stichting voor Fundamenteel Onderzoek der Materie | FOM Projectruimte 11PR2958 | Thomas S Shimizu |

The funders had no role in study design, data collection and interpretation, or the decision to submit the work for publication.

### Author contributions

Johannes M Keegstra, Conceptualization, Resources, Data curation, Software, Formal analysis, Supervision, Investigation, Visualization, Methodology, Writing—original draft, Writing—review and editing; Keita Kamino, Formal analysis, Investigation, Methodology, Writing—review and editing; François Anquez, Data curation, Software, Methodology; Milena D Lazova, Investigation, Methodology, Writing—review and editing; Thierry Emonet, Formal analysis, Funding acquisition, Methodology, Writing—review and editing; Thomas S Shimizu, Conceptualization, Formal analysis, Supervision, Funding acquisition, Investigation, Visualization, Methodology, Writing—original draft, Writing—review and editing

### Author ORCIDs

Johannes M Keegstra (iD) http://orcid.org/0000-0002-8877-4881
Thierry Emonet (iD) http://orcid.org/0000-0002-6746-6564
Thomas S Shimizu (iD) http://orcid.org/0000-0003-0040-7380

Decision letter and Author response
Decision letter https://doi.org/10.7554/eLife.27455.037
Author response https://doi.org/10.7554/eLife.27455.038

## Additional files

**Supplementary files**
• Transparent reporting form
DOI: https://doi.org/10.7554/eLife.27455.035

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
