## [Decision Letter]

Thank you for submitting your article "Generation and attenuation of variability in a bacterial signaling network revealed by single-cell FRET" for consideration by *eLife*. Your article has been reviewed by three peer reviewers, and the evaluation has been overseen by a Reviewing Editor and Naama Barkai as the Senior Editor. The reviewers have opted to remain anonymous.

The reviewers have discussed the reviews with one another and the Reviewing Editor has drafted this decision to help you prepare a revised submission.

Summary:

The authors setup an improved in vivo FRET assay which allowed them to measure intracellular responses at the single cell level. They then applied this unique single-cell technology for studying the cell-cell variability of signaling characteristics of the *E. coli* chemotaxis pathway. By this, they provided convincing evidence that CheB phosphorylation plays a role in suppressing fluctuations, as suggested in previous theoretical work. further, they show that in the absence of methylation-dependent feedback, the combined kinase activity of thousands of receptors follows that of a stochastic two-state switching behavior. This new observation will likely lead to new insights on how receptor clustering affects signaling.

As you can see below, all three reviewers supported the study and would like to see if published in *eLife*, provided that you address the main technical comments that were raised. This includes in particular:

1) Giant FRET activity fluctuations are extraordinary claim and so require more support, at least by providing controls. Why not provide fluorescence sample movies of some individual cells and their segmentation?

2) The interpretation in terms of zeroth-order sensitivity is largely based on the author's own previous modeling. It would therefore be informative to extend the discussion and perhaps include additional papers which might not fit the same framework

3) Fitting of Hill parameters H and K from noisy data might be difficult. please discuss this as requested below.

4) Large coefficients of variation need more explanation/discussion, following the comment below.

Reviewer #1:

The aim of this paper is the predominantly experimental investigation of the variability in signaling in the *E. coli* chemotaxis pathway using in vivo single-cell FRET experiments. Specifically, signaling in single cells is compared with signaling in isogenetic populations. Furthermore, different mutants are considered such as cells expressing only a single receptor type, cells with modified CheB demethylation enzyme, and cells without adaptation (no CheR and CheB enzymes). The latter allows the investigation of signaling noise in absence of gene expression noise, which can be considerable as adaptation enzymes are expressed at a low level. In cells without adaptation very interesting "giant fluctuations" are observed, when the applied ligand concentration is in the sensitive regime of the receptors. Here, apparently thousands of receptors switch in their activity state synchronously. Furthermore, the previously raised idea of zeroth-order sensitivity in adaptation is further promoted, believed to explain the interesting bistable activity profile when no adaptation enzymes are around. The paper is well and clearly written, and full of nice details, and it should be interesting to a broad readership in quantitative biology and systems/synthetic biology. While I do not always fully agree with the interpretation of the results, the questions investigated are certainly well motivated and conceptually well introduced, e.g. the difference in gene and signaling noise, and also why the latter is harder to investigate.

Subsection “Diversity in the ligand response is modulated during population growth”: The extracted Hill coefficients range from approximately 2 to 20 in Figure 2. A Hill coefficient of 20 would be similar to recent data on the motor bias. However, the curves shown in panels B and C show no actual data points, just the fits. Based on Endres et al., (2008) the FRET data for populations expressing receptors from a plasmid varies dramatically from day to day, and in order to extract parameters like the Hill coefficient, a principal component analysis was used, and the uncertainties of the parameters calculated. Here, are the dose-response curves for single cells based on a single time course (a la Figure 2) or is each single cell measured multiple times? If measured only once the curve might be very noisy, and even if measured multiple times, the activity may also vary drastically as shown in Figure 5. How can one reliably estimate Hill coefficients? Certainly, the uncertainty of the parameters should also be estimated. In Figure 2—figure supplement 2 the experimental errors are estimated but I do not fully understand what this figure really shows (basically distributions of the data values are provided but that can already be seen by the scatter plots). Ideally something like the PCA or a Bayesian framework should be applied.

The paper demonstrates that without fully functional CheB phosphorylation feedback, the receptor activity can be bimodal (Figure 3 and text insubsection “CheB phosphorylation feedback attenuates cell-to-cell variability”). To interpret this, the authors reiterate the previous suggestion of zeroth-order sensitivity in adaptation, where enzymes CheR and CheB work at saturation and the activity is highly sensitive to an imbalance in the enzymes. This may lead to strongly fluctuating adaptation times, allowing cells to perform long runs (good in patchy environments, subsection “Diversity in temporal noise: bet-hedging across exploration and exploitation strategies”). Although this idea is quite neat, I have trouble with it. To me, adding noise into a sensory system seems to contradict what is known about other sensory systems. While the authors then say that the phosphorylation feedback turns the bistable activity distribution into a monostable activity distribution, I still have trouble buying the zeroth-order sensitivity argument. In particular, Figure 4 in Clausznitzer et al., (2010) specifically addressed this issue based on FRET data, and no evidence for zeroth-order sensitivity was found (even when CheB phosphorylation is added).

The paper further demonstrates that without adaptation, there can be giant fluctuations in the receptor activity (Figure 5 and text in subsection “Receptor-kinase fluctuations in the absence of adaptation reveal two-level switching”), potentially demonstrating the coordinated switching of thousands of receptors (subsection “Receptor-kinase fluctuations in the absence of adaptation reveal two-level switching”, and subsection “Giant fluctuations and digital switching in adaptation deficient cells”). This is fascinating and unexpected as individual receptor complexes might switch between inactive and active states, but not all complexes in the whole cell together. Can the authors rule out that cells without fully functional CheB and cells with Tsr(QEQE) without adaptation are not in a highly active state, and, while immobilized, tumble a lot, leading to errors in cell segmentation and activity determination?

Reviewer #2:

This manuscript by Keegstra et al. uses improved FRET measurements combined with standard microscopy to quantify various aspects of the chemotaxis systems in *E. coli* cells.

I am not an expert in chemotaxis and therefore defer to reviewers more familiar with that field to assess the novelty of the specific claims. Given the sheer number of results presented it was not possible for me to go back to previous works to judge that in any meaningful way. However, given the authors' great expertise in the area, and the impressive quantity of high-quality work presented, I strongly suspect the paper will be of great interest to that field. I do have some broader concerns but overall, I think the paper still makes a nice contribution and I would recommend it for publication in *eLife*.

My broader concern is unfortunately hard to address. To my taste the authors try to do way too much in one paper. I would have preferred to read a story that drilled down deeper into one or two of the many effects studied. To me the manuscript comes across as a survey of effects, with a somewhat meandering narrative. With a little more work this could likely have been three high-quality papers, each focused on a more specific question. However, it would be unreasonable to expect the authors to address that issue at this stage. They chose to produce a different type of paper, and though I believe that makes it harder for the general reader to take home a clear message (beyond the rather non-distinct message that there is noise in protein networks etc.), it will likely still be important. That said, if in the revisions the authors find that some parts are dispensible, I would encourage them to take those parts out and focus more on the core results. At the very least they may want to tighten the discussion.

Other concerns:

1) It seems the authors went through a substantial effort to optimize the FRET pair, but I wonder if they considered the artefactual effects that can come from the slow and distributed maturation of their mRFP? Even the half-way time of maturation for every red protein I have seen has been longer than 30 minutes. This could have many effects on the measurements. For example, only a fraction of FRET pairs will be active at any given time, that fraction will depend on the history of expression dynamics, and because maturation is distributed, the effect of extrinsic noises can be underestimated because the maturation step will serve as a time-averaging step, much like a low-pass filter. Though the authors obviously cannot be expected to repeat the results with a different FRET pair, this should be discussed.

2) Related to 1, I would like to see a more thorough discussion of potential experimental artefacts. The fact that the average of their single cell data agree with population measurements does not mean that the fluctuations around the average can be trusted. Because this is a novel method I would like to see at least one paragraph describing the controls made to ensure that the noise is due to the biology, and not to imaging (heterogeneity in the evenness of excitation, camera noise etc.), to cell handling (that conditions are uniform in space and time etc.) or to reporter artefacts (that the FRET pair does not affect the circuit's behavior). Only once these controls are completed would it make sense to consider the biology. I do not believe this is a great problem because so much of the data makes sense, but it should still be described for a novel method. On a minor note it should also be pointed out in the main text (my apologies if I just missed it) if the measurements are for total fluorescence or if the authors divide the intensity by the area or volume of the cell. In the former case much of the heterogeneity could come from the differences in cell sizes in a population, so I suspect the authors normalize by cell size.

3) It is striking that the noise is so high in general. The authors compare their results to the extrinsic noise floor of CV=30% in Taniguchi et al. This paragraph needs to be fixed or removed though. The Taniguchi paper considered fluctuations in a cloning strain of *E. coli*, under great starvation conditions, in an unestablished microfluidic device where cells grew twice as slowly as in batch culture with the same medium, and possibly displayed great heterogeneity in growth. When the same protein fusions are imaged under more uniform conditions without using the cloning strain, the total CV drops to <10% in many cases, and the extrinsic part seems to be less than 5%. This pattern has been observed across many labs, including the intrinsic/extrinsic paper by Elowitz and Leibler who reported extrinsic noise levels as low as 5%.

In particular I would like to see more of a discussion for the very high CV observed in the expression levels of CheB, with CVs of 80-90%. Can that be explained extremely weak transcription and strong translation, or do the authors suspect some other mechanism? Can the authors observe noise levels as low as 10% for other genes with the same set-up, to make sure this is not due to the imaging set-up somehow? I noticed the OD was mid-exponential, close to the diaxic shift where some cells start changing gene expression patterns. Could that be the explanation?

4). For the residence time distributions, I would like to see the authors spend more time to confirm that the way residence times are called does not affect the results. I would also make a bigger point of the fact that the residence times actually seem to fit exponential distributions, rather than merely using exponentials to get at the rate constant. For exponentials the

Slightly related to this issue, for other waiting time distributions they see a CV of 20% and call that broad, though the switching times at the end supposedly have CVs of 1, if they are exponential. Though the first waiting times are not from two-state models, it is quite difficult to reduce the heterogeneity in timing even in multi-state switching.

Reviewer #3:

It was a pleasure to read the manuscript by Keegstra el al. on "Generation and attenuation of.…". Indeed, there is nothing to dislike about this paper.

By improving an in vivo FRET assay, which was developed and used to measure the response of a population of cells to external stimuli, the Shimizu group has now for the first time (in my knowledge) been able to measure these intracellular responses at the single cell level. That's a significant achievement all by itself. This unique single-cell measurement, together with existing modeling approach, allows the authors to quantify the cell-cell variability of the intracellular dynamics (response and adaptation) as well as the key components (e.g., Tar/Tsr ratio) in the *E. coli* chemotaxis pathway during different stages (phases) of the population growth process from the response measurements of individual cells. The new technique, single cell FRET, has already shed some new lights onto an old problem/puzzle. In particular, it is shown quite convincingly here that the role of the CheB phosphorylation is to suppress fluctuations in the response regulator as suggested by previous theoretical work. Finally, what is most exciting about the present work is that it revealed something quite unexpected, namely, for at least some of the CheRB- cells (without adaptation) the kinase activity of the entire cell (with thousands of receptors) follows that of a stochastic two-state switching behavior. This new observation will likely lead to new insights on how receptor clustering affects signaling.

The paper is well written. It contains a lot of information, yet it is written in a way easy for the readers to get the main ideas. The paper should be published after the authors consider a few questions/comments that came up while reading the manuscript:

1) Different attractants are used in this paper, e.g. MeAsp for wt cells used in Figure 1 and L-Serine for CheRB- mutants shown in Figure 2. Any reason for doing that, i.e., not using both like what's done in Sourjik and Berg (2004)? From the response data to L-Serine and the model parameters inferred, the response to MeAsp may be predicted, which would be a test for the theoretical model used here.

2) The values of K and H for different individual cells, as shown in Figure 2, are quite interesting. For different cells and cells in different growth phase, the cooperativity H and the inverse sensitivity K seem to collapse onto a single curve. Is there a reason for that? Furthermore, in a previous work ("Lateral density of receptor arrays in the membrane plane influences sensitivity of the *E. coli* chemotaxis response", C. M. Khursigara et al., 2011), it was found that while cooperativity H is higher for cells grown in H1 medium than that of cells grown in TB, its K value is also larger. How does the result shown here relate to that reported in this previous observation?

3) In Figure 4, the blue line shows a very small fluctuation for the ChRB- mutant in comparison with the wt cell (red line). It is a bit confusing in light of the following Figure 5 which shows the giant fluctuation of the CheRB- mutants. The reason for this difference needs to be explained.

4) It needs to be pointed out that the Ornstein-Uhlenbeck equation for kinase activity (Eq. (1) in this manuscript) was first proposed in Tu & Grinstein, (2005), where a 20% fluctuation was given as a lower bound for observing 1-decade of power law CCW duration time distribution. This lower bound is certainly satisfied by the new observation presented in this paper.

---

## [Author Response]

Summary:The authors setup an improved in vivo FRET assay which allowed them to measure intracellular responses at the single cell level. They then applied this unique single-cell technology for studying the cell-cell variability of signaling characteristics of the E. coli chemotaxis pathway. By this, they provided convincing evidence that CheB phosphorylation plays a role in suppressing fluctuations, as suggested in previous theoretical work. further, they show that in the absence of methylation-dependent feedback, the combined kinase activity of thousands of receptors follows that of a stochastic two-state switching behavior. This new observation will likely lead to new insights on how receptor clustering affects signaling.

We thank reviewers #1-3 and the reviewing editor for their time and helpful comments to our manuscript. Before addressing each comment individually, we summarize here the main changes to our manuscript. In response to reviewers’ comments we have made substantial edits to the text (at locations specified within the point-by-point responses below), changes to two of the main figures (Figure 2 and Figure 5), added three new supplementary figures (Figure 1—figure supplement 3, Figure 2—figure supplement 1 and Figure 3—figure supplement 5) and added new panels to two existing supplementary figures (Figure 1—figure supplement 1 and Figure 4—figure supplement 3). Regarding, reviewer #2’s comment that the overall presentation / clarity of narrative could be improved, we have opted not to drastically modify the structure of the text, but have implemented a change in the title of the paper that we feel would help the reader in approaching the rather diverse body of results reported within this paper.

As you can see below, all three reviewers supported the study and would like to see if published in eLife, provided that you address the main technical comments that were raised. This includes in particular:1) Giant FRET activity fluctuations are extraordinary claim and so require more support, at least by providing controls. Why not provide fluorescence sample movies of some individual cells and their segmentation?

We thank the editor for this suggestion. We have added Video 1 which shows fluorescence image sequences for segmented regions of three example cells demonstrating giant step-like fluctuations, together with a synchronously animated time series of the donor, acceptor and ratiometric FRET signal. The image sequences clearly demonstrate that the cells remain immobile during these fluctuations, directly addressing reviewer #1’s concern that the fluctuations might be due to cell movement. We note that within the image sequences, the fluorescence intensity changes appear rather subtle (which is expected for FRET, which typically accounts for a small fraction of total fluorescence intensity), but the time series plots clearly demonstrate that the step-like intensity changes far exceed shot noise levels and furthermore the changes in the donor and acceptor intensities are anti-parallel, which is characteristic of bona fide changes in FRET.

2) The interpretation in terms of zeroth-order sensitivity is largely based on the author's own previous modeling. It would therefore be informative to extend the discussion and perhaps include additional papers which might not fit the same framework

We agree that zero-order ultrasensitivity is not the only possible explanation for the observed bimodality in the steady-state kinase activity (*a*_0_). But it is a particularly simple mechanism capable of generating the observed phenomenology, and thus deserves serious consideration, if as nothing more than a falsifiable hypothesis for more detailed future investigations. Nevertheless, we agree that our presentation could benefit from further clarification of this point, and we have implemented a number of improvements to this end. In the Results section, we have modified the narrative to more generically introduce the nonlinear mapping between the [R]/[B] ratio and the steady-state activity *a*_0_, prior to introducing the possible models that could account for that mapping, including ultrasensitivity (subsection “CheB phosphorylation feedback attenuates cell-to-cell variability”). In the Discussion section, we have added a paragraph discussing the implications and assumptions of the zero-order model, together with other models of nonlinear adaptation kinetics in the literature. In addition, we have furnished a new supplementary figure (Figure 5—figure supplement 4) which demonstrates that models with linear or supralinear dependence of adaptation kinetics on kinase activity (*a*) that have been suggested in the literature cannot on their own provide an explanation for the bimodality in steady-state activity (*a*_0_), while a model with sublinear dependence due to Michalis-Menten kinetics (which can lead to zero-order ultrasensitivity in appropriate parameter regimes) can explain such behavior without invoking any additional assumptions. We once again emphasize that our intention is not to claim that ultrasensitivity is the only possible explanation for the bimodality in *a*_0_, but rather to provide a mechanistic perspective amenable to refinement in future studies.

3) Fitting of Hill parameters H and K from noisy data might be difficult. please discuss this as requested below.

We agree that the reader would benefit from a clearer view of the quality of dose-response data and fitting uncertainties, and thank the editor for this suggestion. We have added a more detailed analysis of Hill parameter estimation uncertainty (including parameter covariance) with a new Figure supplement to Figure 2 and have also modified Figure 2 itself to show representative example of fits to dose response curves.

4) Large coefficients of variation need more explanation/discussion, following the comment below.

We agree that the variability is very large over all, and that the reader would benefit from further comments on this point. To address this, we have added edits to the text (at locations indicated in point-by-point responses below), and in addition furnished a new table within the main text (Table 2), which summarizes the coefficients of variation (CV) of various signaling parameters measured in this work, as well as the temporal variability (*η*), in a manner that facilitates comparison across genotypes and (where available) also to previous estimates from the literature. We have listed each reported CV and show that we have always tried to contrast surprisingly high CV’s with alternative measurements or genotypes in which the CV in the parameter is lower.

Reviewer #1:The aim of this paper is the predominantly experimental investigation of the variability in signaling in the *E. coli chemotaxis pathway using* in vivo single-cell FRET experiments. Specifically, signaling in single cells is compared with signaling in isogenetic populations. Furthermore, different mutants are considered such as cells expressing only a single receptor type, cells with modified CheB demethylation enzyme, and cells without adaptation (no CheR and CheB enzymes). The latter allows the investigation of signaling noise in absence of gene expression noise, which can be considerable as adaptation enzymes are expressed at a low level. In cells without adaptation very interesting "giant fluctuations" are observed, when the applied ligand concentration is in the sensitive regime of the receptors. Here, apparently thousands of receptors switch in their activity state synchronously. Furthermore, the previously raised idea of zeroth-order sensitivity in adaptation is further promoted, believed to explain the interesting bistable activity profile when no adaptation enzymes are around. The paper is well and clearly written, and full of nice details, and it should be interesting to a broad readership in quantitative biology and systems/synthetic biology. While I do not always fully agree with the interpretation of the results, the questions investigated are certainly well motivated and conceptually well introduced, e.g. the difference in gene and signaling noise, and also why the latter is harder to investigate.Subsection “Diversity in the ligand response is modulated during population growth”: The extracted Hill coefficients range from approximately 2 to 20 in Figure 2. A Hill coefficient of 20 would be similar to recent data on the motor bias. However, the curves shown in panels B and C show no actual data points, just the fits. Based on Endres et al., (2008) the FRET data for populations expressing receptors from a plasmid varies dramatically from day to day, and in order to extract parameters like the Hill coefficient, a principal component analysis was used, and the uncertainties of the parameters calculated. Here, are the dose-response curves for single cells based on a single time course (a la Figure 2) or is each single cell measured multiple times? If measured only once the curve might be very noisy, and even if measured multiple times, the activity may also vary drastically as shown in Figure 5. How can one reliably estimate Hill coefficients? Certainly, the uncertainty of the parameters should also be estimated. In Figure 2—figure supplement 2 the experimental errors are estimated but I do not fully understand what this figure really shows (basically distributions of the data values are provided but that can already be seen by the scatter plots). Ideally something like the PCA or a Bayesian framework should be applied.

We thank the referee for this constructive criticism. To make the best use of the stringent photon budget of single-cell FRET experiments, we are deliberately operating near the low signal-to-noise limit, and we couldn’t agree more that fits of nonlinear functions to noisy data need to be interpreted with care. The supplementary figure referred to by the reviewer (formerly Figure 2—figure supplement 2, now Figure 2—figure supplement 3) was indeed provided to address such concerns, and compares the variability in the fitted parameters for experimental data against the that for simulated data in which the only source of variability is added noise representing experimental (shot) noise. More specifically, the grey points in those scatter plots were obtained by adding to the population-averaged dose response curve Gaussian noise of the same magnitude of experimental shot noise and applying the same data fitting procedure applied to the experimental data. In general, the variability of fit parameters for the experimental data exceeds that of the simulated data. The sole exception is the parameter *H* in Tsr+ cells (Figure 2—figure supplement 3), for which the simulations suggest the experimentally observed variability could be dominated by experimental noise.

To better convey the care we have taken in the interpretation of these single-cell dose-response data, and to provide the reader with a means of assessing the fit quality, we have implemented the following improvements:

We have modified Figure 2 to include example fits of the hill function to the single-cell dose response curves. The finite photon budget of single-cell FRET experiments renders multiple dose response measurements on the same set of cells challenging, but at each stimulus the response is averaged over 10-20 data points to average out shot noise. The example fits show that the fit quality is very high, and bad fits are excluded from the analysis.

We have added a supplementary figure (Figure 2—figure supplement 1) showing the parameter uncertainty by decomposing the covariance matrix of the fit of each individual curve into its eigenvectors, and used these to draw error basins in *K-H* space. In this figure, it is shown that indeed the exact value of *H* becomes less constrained when the number of points along the decaying part of the activity to [*L*] decreases, but the values of some cells definitely exceed 10.

Regarding how temporal fluctuations of the type depicted in Figure 5 might affect dose-response measurements of Tsr+ cells, we do not expect a very strong effect for two reasons. First, the timescale of fluctuations we observed in CheRB- cells are very slow (of order 100 s), whereas the ligand-exposure time in our donse-response measurements are relatively short (10-20 s). Temporal fluctuations within the ligand-exposure interval are thus very small. Second, the largest temporal fluctuations are observed in cells with the steepest dose-response curves (expressing Tsr as the sole chemoreceptor), but the large fluctuations are observed only within the very narrow range of ligand concentration corresponding to the dose-response transient (i.e. where activity is neither zero or one). This narrowness of the response regime dominates Hill fits for this genotype, and activity fluctuations within this narrow regime can only mildly affect the fit parameters.

The reviewer also raises a valid concern regarding day-to-day variation, but this is actually much less of an issue for our experiment and analysis. One important difference with the experiments performed in this work and Endres *et al.,* (2008) is that the variability in Figure 2 does not include day-to-day variability because all cells are measured in the same experiment. Furthermore, much of the variability in Endres, 2008 is believed to originate from copy number variation of plasmids from which receptors were expressed, whereas the largest dose-response variability observed in our experiments was in cells with a WT receptor complement, which express receptors from their native chromosomal locus. In our experiments on Tsr+ cells, the receptors were expressed from a plasmid, but in those populations we found cell-to-cell variability to be much lower.

The paper demonstrates that without fully functional CheB phosphorylation feedback, the receptor activity can be bimodal (Figure 3 and text insubsection “CheB phosphorylation feedback attenuates cell-to-cell variability”). To interpret this, the authors reiterate the previous suggestion of zeroth-order sensitivity in adaptation, where enzymes CheR and CheB work at saturation and the activity is highly sensitive to an imbalance in the enzymes. This may lead to strongly fluctuating adaptation times, allowing cells to perform long runs (good in patchy environments, subsection “Diversity in temporal noise: bet-hedging across exploration and exploitation strategies”). Although this idea is quite neat, I have trouble with it. To me, adding noise into a sensory system seems to contradict what is known about other sensory systems.

We agree with the reviewer that generating noise in a sensory system is at first counterintuitive, given that noise in communication channels do generally degrade information transfer. However, while it is possible (and can be meaningful) to view the bacterial chemotaxis signaling pathway as a sensory communication channel, it clearly functions not only to transduce information, but also to modulate motile behavior. Thus, it functions not only as a sensory system but also a locomotor control circuit. This pleiotropy demands that we consider its design in a somewhat different light from the more dedicated sensory systems (typically from more complex organisms) on which we surmise the reviewer’s opinion might be founded.

Viewed as a control circuit implementing a random-walk search strategy, it is entirely natural to expect that selection acts on behavioral statistics, and by extension, also on the intracellular stochastic processes that generate those statistics. The idea that temporal noise in the intracellular CheY-P signal can transform *E. coli*’s run-and-tumble statistics from a Brownian walk into a Levy walk is now very well established with both experimental and theoretical support (Korobkova et al., 2004, Tu & Grinstein, 2005, Emonet & Cluzel, 2008, Matthäus et al., 2011, Zaburdaev et al., 2015), and in addition, it has also been shown theoretically that a finite amount of temporal noise can enhance chemotactic responses in shallow spatial gradients (Sneddon et al., 2012, Flores et al., 2012).

Yet, there must also be ecological contexts in which the reviewer’s opinion/intuition is correct, that is, noise is clearly not always beneficial. Indeed, the same studies cited above (Sneddon et al., 2012, Flores et al., 2012) demonstrated that in steeper gradients, chemotactic performance monotonically deteriorates with noise. Our finding in the present work that the amplitude of temporal noise (*η*) is highly variable from cell to cell in an isogenic population (*CV(η*)≈0.44) thus raises the compelling possibility that *E. coli* hedges its bets by phenotypically diversifying into subpopulations that excel at exploratory behavior (driven by large signal fluctuations) and those that excel in more exploitative behavior (with lower signal fluctuations).

To better convey to the reader these concepts, which can at first be counter-intuitive, we have added the following passage to the Discussion section:

The large temporal noise we observed in wildtype (CheRB+) cells may seem counterintuitive, given that the chemotaxis pathway is a transduction path for sensory information, and noise generally reduces information transmission capacity of communication channels (Shannon, 1949). However, the chemotaxis signaling pathway is not only a sensory system but also a control circuit for motile behavior, and recent studies have highlighted the importance of considering the behavioral context in understanding the design of this signaling pathway (Dufour et al., 2016, Wong et al., 2016, Long et al., 2017).

While the authors then say that the phosphorylation feedback turns the bistable activity distribution into a monostable activity distribution, I still have trouble buying the zeroth-order sensitivity argument. In particular, Figure 4 in Clausznitzer et al., (2010) specifically addressed this issue based on FRET data, and no evidence for zeroth-order sensitivity was found (even when CheB phosphorylation is added).

While we agree with the reviewer that Figure 4 in Clausznitzer et al., (2010) shows no evidence for zeroth-order sensitivity, it shows no evidence *against* zero-order ultrasensitivity either. This is further substantiated in the supporting information of the same paper where the authors show that a model with ultrasensitivity and CheB-P feedback fits the data in Figure 4 nearly as well (χ2=0.0025) as a model without ultrasensitivity (χ2=0.0022). Clonal populations of *E. coli* exhibit substantial cell-to-cell variability in their chemotaxis system (e.g. Spudish and Koshland, 1976; Park et al., 2010; Dufour et al., 2016; and this manuscript). Given that ultrasensitivity manifests itself mostly in the nonlinear adaptation kinetics of single cells, it is not surprising that it is difficult to assess the presence of this effect from FRET measurements averaged over several hundred cells (Figure 4 in Clausznitzer et al.). Note however that even at the population level, temporal ramp-response data (Shimizu et al., 2010) were shown to fit well Michaelis-Menten adaptation kinetics, a functional form that tends to produce fluctuations when the enzymes work at or near saturation.

The main point of Figure 4 from Clausznitzer *et al.* is not about ultrasensitivity but about the nonlinear activation of CheB, which becomes suddenly very high at high level of fractional kinase activity (*a* > 0.7). Clausznitzer *et al.* proposed that even though there is no known molecular mechanism for this effect, one could model it by assuming that the demethylation rate exhibits a supralinear (cubic) dependency with respect to the fractional kinase activity rather than the quadratic dependency that comes from the CheB-P feedback. Following the reviewer’s suggestion we examined (new Figure 3—figure supplement 5) how the steady-state activity (*a*_0_) depends on the CheR/CheB ratio for both sublinear kinetic models (i.e. Michaelis-Menten (de)methylation rates, which can lead to ultrasensitivity) and for the supralinear kinetic models studied by Clausznitzer et al., where the enzyme kinetics are assumed to operate at kinetic order > 1. In both cases we consider what happens when the CheB phosphorylation feedback is present (quadratic or cubic dependencies of the demethylation rate on *a*) or not. In the presence of CheB feedback, the model with Michaelis-Menten rates the steady state activity a0 is more sensitive to cell-to-cell variations in [R]/[B], than the other models as expected. However, when the CheB feedback is removed, the ultrasensitive model shows an increased sensitivity of *a*_0_ to [R]/[B], as seen in our data (CheB-D56E mutant), whereas the model with enzymes operating in the linear regime exhibit a reduced sensitivity of a0 to [R]/[B], independently of the parameter values chosen. To explain bimodality in activity levels due to loss of phosphorylation feedback, a much steeper dependence of *a*_0_ to [R]/[B] is required.

This being said, we do acknowledge that the model with saturated Michaelis-Menten kinetics is a coarse-grained model that can be further refined, especially concerning the interaction of the methylation enzymes with the receptor tether. Also, we do not rule out alternative explanations for bimodality in *a*_0_. Therefore, we have added the following paragraph in the Discussion section:

By analyzing simplified models of adaptation kinetics, we found that a bimodal 𝑃(a_0_) could occur in the absence of phosphorylation feedback if the enzyme kinetics of CheR and CheB depend sublinearly on the activity 𝑎 of their receptor substrates. As a limiting case, when both enzymes work at or near saturation, this model leads to zero-order ultrasensitivity (Goldbeter and Koshland, 1981; Emonet and Cluzel, 2008), which could act as a strongly non-linear transfer function 𝑓([R]/[B]) that converts a unimodal distribution 𝑃([R]/[B]) into a bimodal 𝑃(𝑎0). We note that ultrasensitivity due to sublinear (Michaelis-Menten) enzyme kinetics is by no means the only possible explanation for the observed bimodality in 𝑃(a_0_). Any mechanism that renders f([R]/[B]) a strongly nonlinear (sigmoidal) function could lead to the same effect. The merit of the sublinear kinetic (ultrasensitivity) model is in its simplicity, but it is worth noting that reality is likely to be more complex due to, for example, effects of spatial organization. It is known that both CheR and CheB interact with chemoreceptors not only at their substrate modification residues, but also with a second binding site on a flexible tether at the receptor C-terminus. Such bivalent interactions with the receptor array could affect the movement of these enzymes across the receptor lattice (Levin et al., 2002), and such movements could shift the balance between processivity and distributivity of enzyme activity on their substrate receptors (Pontius et al., 2013), which could in turn attenuate or enhance the nonlinearity in the relationship f([R]/[B]) be- tween the enzyme expression ratio [R]/[B] and the steady-state activity a_0_ of their substrate receptors (Takahashi et al., 2010).

The paper further demonstrates that without adaptation, there can be giant fluctuations in the receptor activity (Figure 5 and text in subsection “Receptor-kinase fluctuations in the absence of adaptation reveal two-level switching”), potentially demonstrating the coordinated switching of thousands of receptors (subsection “Receptor-kinase fluctuations in the absence of adaptation reveal two-level switching”, and subsection “Giant fluctuations and digital switching in adaptation deficient cells”). This is fascinating and unexpected as individual receptor complexes might switch between inactive and active states, but not all complexes in the whole cell together. Can the authors rule out that cells without fully functional CheB and cells with Tsr(QEQE) without adaptation are not in a highly active state, and, while immobilized, tumble a lot, leading to errors in cell segmentation and activity determination?

We appreciate this concern, which seems to be shared by reviewer 2, and in the Results section we emphasize that ratiometric FRET measurements involve an anti-parallel response in the donor and acceptor channels, which can help to discriminate true changes in FRET from various artefacts that can cause changes in the donor/acceptor ratio. We have also added Video 1 which shows cells that do not move during image acquisition. Cells with small response amplitudes, which is a strong indication of low kinase activity but is only a small fraction of the population, are not included in the switching analysis. The response amplitudes of cells with CheB mutants is on average not different from WT CheB, and cells with a very low steady-state activity manifest themselves by barely responding to the addition of attractant, but with a large response to the removal of ligand.

Reviewer #2:This manuscript by Keegstra et al. uses improved FRET measurements combined with standard microscopy to quantify various aspects of the chemotaxis systems in E. coli cells.I am not an expert in chemotaxis and therefore defer to reviewers more familiar with that field to assess the novelty of the specific claims. Given the sheer number of results presented it was not possible for me to go back to previous works to judge that in any meaningful way. However, given the authors' great expertise in the area, and the impressive quantity of high-quality work presented, I strongly suspect the paper will be of great interest to that field. I do have some broader concerns but overall, I think the paper still makes a nice contribution and I would recommend it for publication in eLife.My broader concern is unfortunately hard to address. To my taste the authors try to do way too much in one paper. I would have preferred to read a story that drilled down deeper into one or two of the many effects studied. To me the manuscript comes across as a survey of effects, with a somewhat meandering narrative. With a little more work this could likely have been three high-quality papers, each focused on a more specific question. However, it would be unreasonable to expect the authors to address that issue at this stage. They chose to produce a different type of paper, and though I believe that makes it harder for the general reader to take home a clear message (beyond the rather non-distinct message that there is noise in protein networks etc.), it will likely still be important. That said, if in the revisions the authors find that some parts are dispensible, I would encourage them to take those parts out and focus more on the core results. At the very least they may want to tighten the discussion.

We agree with the reviewer that this paper contains a lot of information. However, we feel that all the experimental results shown are necessary to convey the message that signaling variability in *E. coli* chemotaxis is large and pervasive and that a certain completeness in measured signaling parameters is of added value. However, we have also critically examined the paper text and have almost completely removed two paragraphs in the Discussion Section, which we considered interesting but optional speculation.

Also, to focus the reader on the core results of the paper we have changed the title from “Generation and attenuation of signaling variability in a bacterial signal network (…)” to “Phenotypic diversity and temporal variability in a bacterial signaling network (…)”. We feel that this change to the title goes some way in addressing the reviewer’s concern that the narrative is somewhat meandering, by clarifying from the outset that we have studied here multiple types of variability that fall into two broad classes: (i) ‘phenotypic diversity’, differences across cells in parameters that persist over the cells’ generation time, due largely to gene expression fluctuations, and (ii) temporal variability within each individual cell, due to factors other than gene expression such as protein-protein interactions.

1) It seems the authors went through a substantial effort to optimize the FRET pair, but I wonder if they considered the artefactual effects that can come from the slow and distributed maturation of their mRFP? Even the half-way time of maturation for every red protein I have seen has been longer than 30 minutes. This could have many effects on the measurements. For example, only a fraction of FRET pairs will be active at any given time, that fraction will depend on the history of expression dynamics, and because maturation is distributed, the effect of extrinsic noises can be underestimated because the maturation step will serve as a time-averaging step, much like a low-pass filter. Though the authors obviously cannot be expected to repeat the results with a different FRET pair, this should be discussed.

[We thank the reviewer for raising this issue and we believe confusion might arise from] intuitions from measurements of gene-regulatory circuit dynamics, on which most experimental studies on cellular noise and variability have focused. In such experiments, the dynamics of the circuit often play out on timescales comparable to the maturation time and the signal that provides the circuit dynamics are changes in protein abundance, measured by fluorescence intensity. The intrinsic response time of the signal in such an experiment is limited by the turnover kinetics of the fluorescent species, and a delay in maturation kinetics can influence the dynamics and distributions to be smeared out, as described by the reviewer. However, our FRET experiment on the bacterial chemotaxis system differs from those experiments in the sense that the ratiometric FRET signal measures protein interaction, not abundance, and is far less sensitive to changes in the copy number (due e.g. to maturation) of fluorophores.

More concretely, in our FRET analysis of the chemotaxis network, dynamic changes in intensity due to maturation cannot be strictly excluded, but its effects are negligible in our analysis for three reasons:

(a) Small amplitude: under our experimental conditions, its effects on the net fluorescence intensity drift (and hence also the FRET signal), are very small compared to that of photobleaching.

(b) Timescale separation: the timescales involved in the fluorescence intensity drifts (including both bleaching and maturation) are much longer (> 1hr) than the longest timescales (~400s) we have studied in this paper.

(c) Further attenuation in data analysis: the fluorescence-intensity drift correction we apply in our data analysis means that effects of processes contributing to drift (bleaching, maturation etc) on both the intensity and dynamics of signals are effectively canceled, up to the accuracy of the fits used in the correction.

2) Related to 1, I would like to see a more thorough discussion of potential experimental artefacts. The fact that the average of their single cell data agree with population measurements does not mean that the fluctuations around the average can be trusted. Because this is a novel method I would like to see at least one paragraph describing the controls made to ensure that the noise is due to the biology, and not to imaging (heterogeneity in the evenness of excitation, camera noise etc.), to cell handling (that conditions are uniform in space and time etc.) or to reporter artefacts (that the FRET pair does not affect the circuit's behavior). Only once these controls are completed would it make sense to consider the biology. I do not believe this is a great problem because so much of the data makes sense, but it should still be described for a novel method. On a minor note it should also be pointed out in the main text (my apologies if I just missed it) if the measurements are for total fluorescence or if the authors divide the intensity by the area or volume of the cell. In the former case much of the heterogeneity could come from the differences in cell sizes in a population, so I suspect the authors normalize by cell size.

We appreciate the reviewer’s rigor concerning this point. We agree that the reader benefits from a more elaborate discussion of potential artefacts. Hence to address these concerns we have changed a paragraph in the Results section to explain more clearly how the FRET experiment is performed, and in which we stress the anti-parallel response signature of the ratiometric FRET imaging:

“A field of *E. coli* cells expressing this FRET pair were immobilized on a glass surface imaged in two fluorescence channels, and segmented offline to obtain fluorescence intensities of donor and acceptor. From the fluorescence ratio FRET time series for each cell in the field of view (see Materials and methods section) can be computed, after dividing out the decay (Figure 1—figure supplement 1) in each channel due to bleaching. Ratiometric FRET provides an anti-parallel response signature and confers robustness to parallel fluctuations that affect both fluorescent channels, such as differences in absolute fluorescence intensity due to inhomogeneous illumination and differences in cell size.”

We have also changed and added the following figures to show that the cellular variability cannot be explained by experimental parameters and that the FRET protein fusions generate a fully functional genotype:

A supplementary figure to Figure 1 (Figure 1—figure supplement 3) which shows the correlations between the signaling parameters (parameters recovery time, steady-state activity and adaptation precision) and experimental parameters. It clearly shows the variability in signaling cannot be explained by imaging parameters such as illumination intensity or fluorophore emission. The only significant correlation we found between fluorophore intensity and recovery time is weak (-0.3) and the recovery time does not correlate significantly with the distance from the highest intensity on the sample plane.

An extra panel to Figure 1—figure supplement 1, which shows that cells expressing the FRET plasmid are fully functional in chemotaxis. The fact that their collective behavior agrees well with wild-type, provides also some evidence that not only the means of certain signaling parameters can be trusted, but also the distributions of those parameters from cell to cell. We do note that this functional phenotype is a sufficient but not necessary condition for ruling out fluorophore interference with the kinase activity, since downstream artefacts cannot disqualify the measurement of upstream pathway activity.

We have added panels to show how reproducible the signaling parameter distributions are, for Figure 1, Figure 2—figure supplement 1 and Figure 4—figure supplement 3. Generally, they show very high agreement between independent experiments.

Finally, in the Discussion section we have added some more general statements regarding the strength and limitations of the method:

"The anti-parallel response signature of ratiometric FRET provides a good way to discriminate genuine FRET changes from imaging artifacts. As in population-level FRET, single-cell FRET is most easily applied to study large and rapid changes in signaling (e.g. response to step stimuli), but we have shown that with careful correction of drifts in the signal level (primarily due to bleaching), it can be applied effectively to measure more subtle changes in signaling over extended times, including steady-state fluctuations. Nevertheless, absolute quantification of single-cell variability remains challenging, since any experimental noise source can potentially contribute to the observed variability. Although we have not undertaken here a comprehensive survey of experimental noise sources for single-cell FRET, our results demonstrate meaningful differences in variability across cells measured under identical experimental conditions. The experimental duration for single-cell FRET is photon-limited, meaning that optimal experimental strategies must carefully negotiate with a finite photon budget an inherent trade-off between measurement duration, temporal resolution, and signal-to noise ratio. Future improvements of donor/acceptor fluorophores (in parameters such as photostability, brightness, maturation, as well as FRET efficiency) could enhance the effective photon budget, and hence the power of the experiment.

3) It is striking that the noise is so high in general. The authors compare their results to the extrinsic noise floor of CV=30% in Taniguchi et al. This paragraph needs to be fixed or removed though. The Taniguchi paper considered fluctuations in a cloning strain of E. coli, under great starvation conditions, in an unestablished microfluidic device where cells grew twice as slowly as in batch culture with the same medium, and possibly displayed great heterogeneity in growth. When the same protein fusions are imaged under more uniform conditions without using the cloning strain, the total CV drops to <10% in many cases, and the extrinsic part seems to be less than 5%. This pattern has been observed across many labs, including the intrinsic/extrinsic paper by Elowitz and Leibler who reported extrinsic noise levels as low as 5%.

We agree that the CV of many of the parameters we extract suggest that the underlying variability in the expression of chemotaxis genes is greater than the 5-10% lower limit. Relatively large variability in the expression of chemotaxis genes is consistent with published flow cytometry measurements of chemotaxis genes by Kollmann et al. (2005) who reported a CV of 0.67 for the *cheY* gene. For the *cheA-cheY* gene pair they reported η_int_ = 0.26 and η_ext_ = 0.35 for the intrinsic and extrinsic components of noise, respectively. Yoney and Salman, (2015) report a Tar/Tsr ratio distribution with a CV≈0.5 and a Pearson correlation coefficient between the *tar* and *tsr* genes of ≈0.65. These values from the published literature on expression variation of chemotaxis genes are compatible with our finding that strong cell-to-cell variability is pervasive in *E. coli* chemotaxis signaling parameters, and underscore our interpretation that the large CVs likely reflect that of the underlying variability in gene expression.

Regarding our reference to the work of Taniguchi et al., this was intended to support the idea that in general a considerable fraction of the CV can be attributed to extrinsic noise (correlated variation), which is relevant for considering effects on the chemotaxis system, where it has been shown (e.g. in Kollmann et al., 2005) that many properties of the system depend on the ratio between expression levels (e.g. CheR/CheB, Tar/Tsr, CheR/receptor), rather than the expression level of individual components. Observing a lower limit to the CV (which scales as the inverse square root of the mean at low expression levels where intrinsic noise dominates), Taniguchi et al. attributed that lower limit to a noise floor due to extrinsic noise. However, we do appreciate the reviewer’s reservations about this interpretation by Taniguchi et al., and have thus removed the reference to that work. Commenting on such genome-wide gene expression trends is in any case somewhat superfluous in this context, given that a high degree of covariation among the expression level of chemotaxis genes has been experimentally demonstrated both at the single-cell (Kollmann et al., 2005) and population levels (Li and Hazelbauer, 2004), and is also expected given the architecture of the flagellar regulon in which all chemotaxis genes are under the control of a common master regulator (Chilcott and Hughes, 2000). These observations underscore the significance of our finding that the uncorrelated component of variability in the Tar/Tsr ratio is much higher than expected. We have included these arguments above in the Results section.

More generally, we believe reviewer #2’s concern here is related to his/her point 2, namely that the high CVs we observe might reflect experimental artifacts rather than *bona fide* biological variability. We agree with reviewer #2 that experimental results with a lower level of variability would provide proof that the observed variability is not simply the result of hitting a lower bound arising from experimental limitations. In our case, we can contrast high levels of variability with experiments in which the variability is much lower. In case we cannot provide such an experiment, we compare the variability directly to the variability of relevant parameters reported in the literature. This is summarized below.

Adaptation time/precision:The CV for the recovery time is around 0.2. This quantity has been estimated in previous studies by measuring recovery times of motor switching bias after ligand stimulation. In one study, (Berg and Tedesco, 1975) found a large variability (CV≈0.48, 17 cells). Another study (Spudich and Koshland, 1976) reported a lower value (CV≈0.2), which is in agreement with a more recent study (Min *et al.,* 2009, CV≈0.18, 496 pairs) that observed tumbles in optically trapped bacteria. While these results might not be directly comparable due to differences in experimental conditions and analysis, our estimate (CV≈0.2) certainly falls within the reported range. The CV we find for the imprecision of adaptation is relatively high (0.4). Adaptation imprecision has only started to receive attention fairly recently, and to our knowledge, no other study has quantified it at the single-cell level. However, a high CV for the adaptation precision can be rationalized by the hypothesis that this precision is mainly determined by the expression-level ratios among the multiple chemoreceptor species (Meir et al., 2010; Neumann et al., 2014), combined with our finding that the variability of those ratios are very high (see below).

Dose-response parameters:For ligand dose-response curves we find high variability in Hill-fit parameters (*K* and *H*). The inverse sensitivity (*K*) is in general better constrained by the data than the coopertivity (H), as illustrated in the newly furnished Figure 2 Supp. 1b, with the median of the uncertainty (defined as the 95% confidence interval width) divided by the fitted parameter value being 0.19. The variability in *K* for cells with all five chemoreceptors is quite high (CV=0.49), but a much lower variability for the same parameter is observed in cells expressing Tsr as the sole chemoreceptor species (CV=0.16).

Steady-state activity:In our experiments comparing different network topologies due to CheB phosphorylation feedback, we find very high CVs for the steady-state activity *a*_0_ in cells deficient in phosphorylation feedback (1.10 and 1.07 for respectively CheBc and CheB-D56E) while the CV for cells expressing wild-type CheB is considerably lower (0.7). The latter variability for wild-type CheB is still quite large in absolute terms, but we note that in these cells CheB is expressed from a plasmid, which is expected to increase cell-to-cell variability due to plasmid copy number variation, and also by breaking the translational coupling between CheR and CheB (Løvdok et al., 2009). In the case of wild-type cells (which express both CheR and CheB from their native polycistronic operon) the variability in *a*_0_ is much lower (CV=0.23) as shown in Figure 1 (although we note this comparison with the data of Figure 1 requires caution, as the overall noise-to-signal ratio was also lower in the latter experiment due to differences in experimental conditions).

Temporal fluctuations: The temporal noise amplitudes of the steady-state FRET signal in wild-type cells is much larger (0.44) than cells lacking genes for the adaptation enzymes CheR and CheB (0.09). The CV for the cell-to-cell variability in those noise magnitudes across wild-type cells is 0.6. To compare the variability in noise levels between CheRB+ and CheRB- we find 0.55 and 1.25, respectively. We not that the latter value is high because of a low mean noise amplitude. Without normalization, the variability (σ) of RB+ is 0.24 compared to 0.11 for RB-.

Gene expression variability:We measure a CV of 0.8-0.9 for the expression of CheB. The expression level of CheZ and CheY (Figure 3—figure supplement 2) is much less noisy (CV=0.3). We believe both have contributions from plasmid copy number variation. For these non-ratiometric measurements we do correct for cell size (segmentation surface area) and inhomogeneous illumination. We have added clarifying statements about these corrections to the figure captions in Figure 3—figure supplement 1 and Figure 3—figure supplement 2. We have not measured the gene expression variability of the Tar and Tsr proteins, but using a model on the FRET data, we extract a CV of 0.5 in the Tar/Tsr ratio, which is indeed surprisingly high given the high expression level of these proteins (>10^3^ copies per cell), but as noted in the manuscript text it is in good agreement with published dual reporter flow cytometry experiments (CV=0.45, Yoney and Salman, 2005).

While all of the above information is present in the main text and figures we agree that a brief summary of variability (both across cells within a population, over time within a single cell) would aid the reader in making these comparisons. Hence, we have furnished a new table (Table 2) to the paper in which our main findings are summarized which is referred to in the Discussion Section.

In particular I would like to see more of a discussion for the very high CV observed in the expression levels of CheB, with CVs of 80-90%. Can that be explained extremely weak transcription and strong translation, or do the authors suspect some other mechanism? Can the authors observe noise levels as low as 10% for other genes with the same set-up, to make sure this is not due to the imaging set-up somehow? I noticed the OD was mid-exponential, close to the diaxic shift where some cells start changing gene expression patterns. Could that be the explanation?

We agree with the reviewer that the variability in CheB expression is high. But this is not unexpected given that measured protein copy numbers for CheB (and also CheR) in wildtype cells are known to be very low (few hundred copies per cell; Li and Hazelbauer, 2004), unlike most other chemotaxis proteins which are expressed at much higher levels (thousands of copies per cell). It is certainly plausible that the high CVs we observe for CheB expression reflects weak transcription coupled with strong translation (which would enhance noise due to the resulting ‘burstiness’ in protein synthesis), but our data do not on their own argue for or against this particular mechanism. We note that CheB expression in these experiments are from plasmids and this may cause additional cell-to-cell variability due to plasmid copy number variation. Hence, we do not make any claims regarding the absolute level of CheB expression variability in wildtype cells, but have merely made the observation that the variability among the three CheB mutants is similar under our experimental conditions. To clarify this issue, we have added a short cautionary statement in the Results section that the CVs measured for CheB likely include a contribution from plasmid copy number variation (subsection “CheB phosphorylation feedback attenuates cell-to-cell variability”):

4). For the residence time distributions, I would like to see the authors spend more time to confirm that the way residence times are called does not affect the results. I would also make a bigger point of the fact that the residence times actually seem to fit exponential distributions, rather than merely using exponentials to get at the rate constant. For exponentials the

Although this comment seems to have been truncated prematurely (the last sentence is incomplete), the reviewer raises the important point that in considering residence-time statistics, not only the mean but also the shape of the distribution matters. We could not agree more with this point, but precisely determining the shape of the residence-time distribution is a significant experimental challenge, given that these time intervals are quite long (mean ~10^2^ s) and a typical single-cell FRET experiment (the duration of which is limited by the finite photon budget to ~10^3^ s) yields at most a few dozen events per cell. We have thus chosen to focus here on an analysis of the mean residence time (and its dependence on each cell’s activity bias), deferring a more detailed investigation of residence time statistics to a future study.

We point out, however, that we have made no explicit assumptions about the shape of the residence-time distribution (exponential or otherwise) in our analysis. Rather, we have shown that, to a first approximation, the mean residence times (*τ*_up_, *t*_down_) scale exponentially (Eq. 3) with the apparent free-energy difference between active and inactive states (*ΔG* = *k_B_T* ln[(1-*a*_1/2_)/*a*_1/2_]), with slopes (*γ*_up,down_) of opposing sign. To clarify this point, we have added the following sentence in the Results section:

“The fact that the mean residence times (τ_up_,τ_down_) scale exponentially with the apparent free energy difference (ΔG) indicates that, to a first approximation, receptor-kinase switching can be treated as a barrier-crossing process.”

Regarding how residence times are called, because the number of switching events in each two-state time series is rather small (typically less than 10 transitions per cell) we chose here to read off transition time points by eye (with ≈ 2.5 s accuracy). Because the amplitude of such transitions far exceeds that of high-frequency (shot) noise, these events can be identified unambiguously by eye, but the gradual drift of the FRET signal baseline (due to bleaching etc.) renders automatic detection challenging. We are developing a more refined (and automated) analysis for a follow-up study that addresses residence-time statistics in more detail, but for our purpose here of analyzing mean residence times (on a logarithmic scale), small errors in residence time intervals have negligible effects.

Slightly related to this issue, for other waiting time distributions they see a CV of 20% and call that broad, though the switching times at the end supposedly have CVs of 1, if they are exponential. Though the first waiting times are not from two-state models, it is quite difficult to reduce the heterogeneity in timing even in multi-state switching.

We agree with the reviewer that whether a CV is large or small depends on the (assumed) distribution underlying the statistics. In the case of the residence times, since the number of events per cell is so small we have not been able to confirm that the underlying distribution is exponential. We have only shown that the average residence time for each cell scales exponentially with the activity-biasing free energy *ΔG*.

While for a passive (thermally driven) process waiting time distributions are expected to be exponential, for the two-level switching behavior we have observed in RB- cells, at this stage we cannot rule out the involvement of one or more active (dissipative) process(es), which can lead to peaked distributions (and hence lower CV’s). In our discussion text we refer to the case of allosteric switching events in flagellar motor rotation, for which recently peaked residence times were measured and interpreted using a non-equilibrium switching model (Wang et al., 2017).

We assume that the “other waiting time distributions” referred to by the reviewer is the recovery-time distribution upon adaptation to a large step stimulus (Figure 1), as that is the only measured time interval for which we report a CV of ≈0.2. A CV of 0.2 is indeed small compared to that of an exponential distribution, but is much greater than mechanistically expected from the underlying stochastic chemical kinetics alone. This basic reason is that adaptational recovery from large step stimuli requires a large number of receptor-methylation events. From previous calibration measurements of an allosteric model of receptor signaling (Shimizu et al., 2010, Figure 7) we can estimate precisely that adaptation to the 500 μM MeAsp stimulus of Figure 1 requires ~1.5 methyl groups per receptor, and there are >10^3^ receptor molecules per cell. It is reasonable to assume that the waiting time for each individual methylation event obeys Poisson statistics (with CV=1), but the recovery time would then be the sum of thousands such independent random variables. So by the central limit theorem the CV would be reduced by the square-root of the number of events, i.e. given *N*_events_>10^3^, the expected CV=(1/ *N*_events_)^1/2^<≈3%. We therefore maintain that the distribution of recovery times we observe upon adaptation to large attractant steps (Figure 1) are substantially broader than expected from the underlying stochastic methylation kinetics alone. To clarify this issue, we have added the following passage to the Results section reporting this CV:

“The variation is also substantial in 𝜏_recovery_ (Figure 1, 𝐶𝑉 =0.20), considering that the underlying kinetics of receptor methylation (catalyzed by CheR) involve thousands of events per cell, but falls within the range.”

Reviewer #3:It was a pleasure to read the manuscript by Keegstra el al on "Generation and attenuation of.…". Indeed, there is nothing to dislike about this paper.
*By improving an* in vivo FRET assay, which was developed and used to measure the response of a population of cells to external stimuli, the Shimizu group has now for the first time (in my knowledge) been able to measure these intracellular responses at the single cell level. That's a significant achievement all by itself. This unique single-cell measurement, together with existing modeling approach, allows the authors to quantify the cell-cell variability of the intracellular dynamics (response and adaptation) as well as the key components (e.g., Tar/Tsr ratio) in the E. coli chemotaxis pathway during different stages (phases) of the population growth process from the response measurements of individual cells. The new technique, single cell FRET, has already shed some new lights onto an old problem/puzzle. In particular, it is shown quite convincingly here that the role of the CheB phosphorylation is to suppress fluctuations in the response regulator as suggested by previous theoretical work. Finally, what is most exciting about the present work is that it revealed something quite unexpected, namely, for at least some of the CheRB- cells (without adaptation) the kinase activity of the entire cell (with thousands of receptors) follows that of a stochastic two-state switching behavior. This new observation will likely lead to new insights on how receptor clustering affects signaling.The paper is well written. It contains a lot of information, yet it is written in a way easy for the readers to get the main ideas. The paper should be published after the authors consider a few questions/comments that came up while reading the manuscript:1) Different attractants are used in this paper, e.g., MeAsp for wt cells used in Figure 1-Serine for CheRB- mutants shown in Figure 2. Any reason for doing that, i.e., not using both like what's done in Sourjik and Berg (2004)? From the response data to L-Serine and the model parameters inferred, the response to MeAsp may be predicted, which would be a test for the theoretical model used here.

In our single-cell FRET experiments, we were mainly concerned with ensuring response saturation for each individual cell. We know that the population averaged FRET signal of RB- cells can be shut down completely with L-serine, but not with MeAsp (Sourjik and Berg, 2002). Otherwise, we know that the response to MeAsp in RB+ shows near-perfect adaptation but to serine the system adapts much less precisely. Hence, we do think that these are nice suggestions for additional experiments and analysis but fall beyond the scope of this work.

2) The values of K and H for different individual cells, as shown in Figure 2, are quite interesting. For different cells and cells in different growth phase, the cooperativity H and the inverse sensitivity K seem to collapse onto a single curve. Is there a reason for that?

We agree that the parameters *K* and *H* of different cells to first approximation seem to collapse on the same curve, while we also note that there is additional variability.

With the two-receptor-species MWC model of Mello and Tu, (2005) one can obtain a relation between *K* and *H* roughly similar to the trend between the *K* and *H* parameters by only changes the relative abundance of the two chemoreceptor species. Hence according to this model the dominant source of variability from cell to cell is the variability in the abundance ratio of the two chemoreceptors Tar and Tsr. The fact that cells from different harvesting OD’s preserve the relation between *K* and *H* also is consistent with our interpretation that the dominant source of variation caused by harvesting OD is the Tar/Tsr ratio.

However, we do note that the model does not provide a direct relation between the abundance of the chemoreceptor and the signaling team size of each chemoreceptor, only the notion that the ratio of the signaling team sizes should be determined by the ratio of expression levels, if one assumes the cluster is well-mixed. Hence inferring absolute abundances of chemoreceptors directly (instead of their ratio) is not possible unless further assumptions are made.

Furthermore, in a previous work ("Lateral density of receptor arrays in the membrane plane influences sensitivity of the E. coli chemotaxis response", C. M. Khursigara et al., 2011), it was found that while cooperativity H is higher for cells grown in H1 medium than that of cells grown in TB, its K value is also larger. How does the result shown here relate to that reported in this previous observation?

The study by Khursigara et al., (2011) mentioned by the reviewer is very interesting and we have added a reference to this work in the Discussion section. However, we have chosen not to include an in-depth comparison of the mentioned work with our results because of experimental differences.

The work of Khursigara et al., (2011) shows that chemoreceptor array structure is hard to predict from expression levels only. While it is known that cells grown in H1 express many more receptors compared to cells grown in TB, the cluster sizes are comparable between the two growth conditions, while the density of the chemoreceptor arrays does change, with H1 cells having more dense arrays. This roughly agrees with the idea that higher chemoreceptor expression increases the signaling team size and hence cooperativity of the response, but we lack the predictive power to relate the signaling team size to the expression level. Hence if there is significant additional variability beyond the Tar/Tsr expression level ratio it is not surprising that the observed anti-correlation between K and H no longer holds.

We do note that the results of the paper are mostly based on population-averaged measurements. Cell-to-cell variability may influence the apparent cooperativity of the population response (see the family of dose response curves in Figure 2 of our manuscript). Hence it is technically possible that the lower cooperativity of cells grown in TB is caused by larger variability. This is consistent with the fact that the referred manuscript shows lower variability in the receptor array density for cells grown in TB (Figure 3), although the number of cells presented is small. We think performing single-cell FRET experiments to study the relation between growth condition and signalling parameters provides a good direction for further studies.

3) In Figure 4, the blue line shows a very small fluctuation for the ChRB- mutant in comparison with the wt cell (red line). It is a bit confusing in light of the following Figure 5 which shows the giant fluctuation of the CheRB- mutants. The reason for this difference needs to be explained.

We agree with the reviewer that the transition from Figure 4 to Figure 5 could be improved. Therefore, we have added two panels to Figure 5, in which the result is shown of an experiment with CheRB- cells with the same genotype as Figure 4, but then with the addition of ligand to bring the CheRB- cells into a sensitive regime. The spectrum shows elevation at low frequencies compared to the case where no ligand is applied. We also included a power spectrum from experiments with CheRB- and only one chemoreceptor, which reveals even more temporal noise. We believe that this additional data will help the reader to see how the fluctuations presented in Figure 4 and Figure 5 are connected.

4) It needs to be pointed out that the Ornstein-Uhlenbeck equation for kinase activity (Eq. (1) in this manuscript) was first proposed in Tu & Grinstein, (2005), where a 20% fluctuation was given as a lower bound for observing 1-decade of power law CCW duration time distribution. This lower bound is certainly satisfied by the new observation presented in this paper.

We have added this reference at Eqn 1 and we have changed another mention of this work in the Discussion section to (subsection “Diversity in temporal variability: bet-hedging across exploration and exploitation strategies”):

"Another theoretical study of the motor noise (Tu and Grinstein 2005), had predicted a more modest noise level of intracellular noise, with a lower bound of 20% of the mean."